# On Differentially Private Subspace Estimation in a Distribution-Free Setting

**Eliad Tsfadia**
Department of Computer Science
Georgetown University
eliadtsfadia@gmail.com

## Abstract

Private data analysis faces a significant challenge known as the curse of dimensionality, leading to increased costs. However, many datasets possess an inherent low-dimensional structure. For instance, during optimization via gradient descent, the gradients frequently reside near a low-dimensional subspace. If the low-dimensional structure could be privately identified using a small amount of points, we could avoid paying for the high ambient dimension.

On the negative side, Dwork et al. [2014] proved that privately estimating subspaces, in general, requires an amount of points that has a polynomial dependency on the dimension. However, their bounds do not rule out the possibility to reduce the number of points for "easy" instances. Yet, providing a measure that captures how much a given dataset is "easy" for this task turns out to be challenging, and was not properly addressed in prior works.

Inspired by the work of Singhal and Steinke [2021], we provide the first measures that quantify "easiness" as a function of multiplicative singular-value gaps in the input dataset, and support them with new upper and lower bounds. In particular, our results determine the first types of gaps that are sufficient and necessary for estimating a subspace with an amount of points that is independent of the dimension. Furthermore, we realize our upper bounds using a practical algorithm and demonstrate its advantage in high-dimensional regimes compared to prior approaches.

## 1 Introduction

Differentially private (DP) Dwork et al. [2006b] algorithms typically exhibit a significant dependence on the dimensionality of their input, as their error or sample complexity tends to grow polynomially with the dimension. This cost of dimensionality is inherent in many problems, as Bun et al. [2014], Steinke and Ullman [2017], Dwork et al. [2015] showed that any method that achieves lower error rates is vulnerable to tracing attacks (also known as, membership inference attacks). However, these lower bounds consider algorithms that guarantee accuracy for worst-case inputs and do not rule out the possibility of achieving higher accuracy for "easy" instances.

**Example: DP averaging.** As a simple prototypical example, consider the task of DP averaging. In this task, the input dataset consists of $d$-dimensional points $x_1, \ldots, x_n \in \mathbb{R}^d$, and the goal is to estimate their average $\frac{1}{n}\sum_{i=1}^{n} x_i$ using a DP algorithm while minimizing the $\ell_2$ additive error. One natural way to capture input "easiness" for this task is via the maximal $\ell_2$ distance between any two points (i.e., points that are closer to each other are considered "easier"). Indeed, Tsfadia et al. [2022], Peter et al. [2024] showed that if the points are $\gamma$-close to each other, and we aim for an accuracy of $\lambda\gamma$ (i.e., an accuracy that is *proportional* to the "easiness" parameter $\gamma$), then it is sufficient and

38th Conference on Neural Information Processing Systems (NeurIPS 2024).

necessary to use $n = \tilde{\Theta}(\sqrt{d}/\lambda)$ points. Equivalently, if we aim for an accuracy of $\alpha$, then by applying these results with $\lambda = \alpha/\gamma$, we obtain that the answer is $n = \tilde{\Theta}(\gamma\sqrt{d}/\alpha)$. This, in particular, implies that when $\gamma \leq \alpha/\sqrt{d}$ (i.e., the points are very close to each other), then $\tilde{O}(1)$ points are sufficient, but for $\gamma = \alpha/d^{1/2-\Omega(1)}$, a polynomial dependency on $d$ is necessary in general.

**DP subspace estimation.** In this work, we consider the more complex problem of DP subspace estimation: Given a dataset $X = (x_1, \ldots, x_n) \in (\mathbb{R}^d)^n$ of unit norm points and a parameter $k$, estimate the top-$k$ subspace of $Span\{x_1, \ldots, x_n\}$. The main goal of this work is to answer the following meta question:

**Question 1.1.** *How should we quantify how "easy" a given dataset is for DP subspace estimation?*

Since the dimension $d$ is very large in many settings, we aim at providing tight measures that smoothly eliminate the dependency on $d$ as a function of input "easiness". In particular, we want to be able to identify when we can avoid paying on the ambient dimension $d$, and when a polynomial dependency on $d$ is unavoidable.

## 1.1 Motivation: DP-SGD

To motivate the problem, consider the task of privately training large neural networks. The most commonly used tool to perform such a private training is the differentially-private stochastic gradient descent (DP-SGD) Abadi et al. [2016b], Bassily et al. [2014], Song et al. [2013] – a private variant of SGD that perturbs each gradient update with random noise vector drawn from an isotropic Gaussian distribution. However, this approach does not differentiate between "easy" gradients and "hard" ones, which results with high error rates when the ambient dimension - the number of parameters in the model - is large. However, empirical evidence and theoretical analysis indicate that while training some deep learning models, the gradients tend to live near a low-dimensional subspace Abadi et al. [2016a], Li et al. [2018b], Gur-Ari et al. [2018], Li et al. [2018a], Demeniconi and Chawla [2020], Zhou et al. [2021], Feng and Tu [2020], Li et al. [2022], Golatkar et al. [2022], Kairouz et al. [2020]. In particular, Gur-Ari et al. [2018] showed that in some cases, the low dimension is the number of classes in the dataset, and the gradients tend to be close and well-spread inside this subspace. If we could exploit such a low-dimensional structure into an (inexpensive) private and useful projection matrix, we could reduce the error of DP-SGD by making it dependent solely on the low dimension.

We start by defining the setting of DP subspace estimation more formally.

## 1.2 Subspace Estimation

We consider the setting of Dwork et al. [2014]. That is, the input dataset consists of $n$ points of unit norm $x_1, \ldots, x_n \in \mathcal{S}_d := \{v \in \mathbb{R}^d : \|v\|_2 = 1\}$ and a parameter $k$, and the goal is to output a $k$-dimensional projection matrix $\Pi$ such that $\Pi \cdot X^T$ is "close" to $X^T$ as possible, where $X$ denotes the $n \times d$ matrix whose rows are $x_1, \ldots, x_n$. We measure the accuracy of our estimation using the "usefulness" definition of Dwork et al. [2014]:

**Definition 1.2** ($\alpha$-useful). *We say that a rank-$k$ projection matrix $\Pi$ is $\alpha$-useful for a matrix $X \in (\mathcal{S}_d)^n$ if for any $k$-rank projection matrix $\Pi'$:*

$$\left\|\Pi \cdot X^T\right\|_F^2 \geq \left\|\Pi' \cdot X^T\right\|_F^2 - \alpha \cdot n,$$

*where $\|\cdot\|_F$ denotes the Frobenius norm.*[1]

Observe that any projection matrix is 1-useful for any $X$ (because $\|X\|_F^2 = \sum_{i=1}^n \|x_i\|_2^2 = n$). Therefore, we will be interested in smaller values of $\alpha$ (e.g., 0.001).

## 1.3 Prior Works

Without privacy restrictions, we can find a 0-useful (i.e., optimal) solution using Singular-Value Decomposition (SVD). The SVD of $X$ is $X = U\Sigma V^T$, where $U \in \mathbb{R}^{n \times n}$ and $V \in \mathbb{R}^{d \times d}$ are unitary

---

[1] The Frobenius norm of a matrix $A = (a_i^j)_{i \in [n], j \in [d]}$ is defined by $\|A\|_F = \sqrt{\sum_{i \in [n], j \in [d]} (a_i^j)^2}$.

matrices, and $\Sigma$ is an $n \times d$ diagonal matrix which has values $\sigma_1 \geq \ldots \geq \sigma_{\min\{n,d\}} \geq 0$ along the diagonal. The top-$k$ rows subspace of $X$ is given by the span of the first $k$ columns of $V$, and it can be computed, e.g., by applying Principal Component Analysis (PCA) on the covariance matrix $A = X^T X$ (the eigenvectors of $A$ are the columns of $V$).

With differential privacy, however, the problem is much harder, and Dwork et al. [2014] showed a lower bound of $n \geq \tilde{\Omega}(k\sqrt{d})$ for computing a 0.001-useful $k$-rank projection matrix under $(1, \Omega(1/n^2))$-DP. This bound, however, only holds for algorithms that provide accuracy for worst-case instances and does not rule out the possibility of achieving high accuracy with smaller values of $n$ for input points that are very close to being in a $k$-dimensional subspace (i.e., "easy" instances).

Perhaps the easiest instances are those that *exactly* lie in a $k$-dimensional subspace, and are well-spread within it (i.e., there is no $(k-1)$-dimensional subspace that contains many points). Indeed, Singhal and Steinke [2021] and Ashtiani and Liaw [2022] developed $(\varepsilon, \delta)$-DP algorithms for such instances that precisely recover the subspace using only $n = \tilde{\Theta}_{\varepsilon,\delta}(k)$ points. However, while these algorithms are robust to changing a few points, they are very brittle if we change all the points by a little bit.

One approach to smoothly quantify how much a dataset is "easy" is to consider the *additive-gap* $\sigma_k^2 - \sigma_{k+1}^2$. Indeed, Dwork et al. [2014], Gonem and Gilad-Bachrach [2018] present $(\varepsilon, \delta)$-DP algorithms that output 0.001-useful projection using $n = \tilde{\Theta}_{\varepsilon,\delta}\left(\frac{k\sqrt{d}}{\sigma_k^2 - \sigma_{k+1}^2}\right)$ points. Yet, the downside of such additive-gap based approaches is their inherent dependency on the dimension $d$. Even in the extreme case where the points exactly lie in a $k$-dimensional subspace and well-spread within it, the additive gap $\sigma_k^2 - \sigma_{k+1}^2$ is at most $n/k$, which still results with a polynomial dependency on $d$.

The only existing approach to eliminate the dependency on $d$ in some non-exact cases is the one of Singhal and Steinke [2021] (their "approximate" case). Rather than quantifying easiness as a function of the input dataset, they consider a setting where the points are sampled i.i.d. from some distribution, and implicitly measure how "easy" a distribution is according to some stability notion. In particular, they show that a $d$-dimensional Gaussian $\mathcal{N}(\vec{0}, \Sigma)$ with a *multiplicative-gap* $\frac{\sigma_{k+1}(\Sigma)}{\sigma_k(\Sigma)} \leq \tilde{\Theta}_{\varepsilon,\delta,k}\left(\frac{1}{d^2}\right)$ is "stable" enough for estimating the top-$k$ subspace of $\Sigma$ with sample complexity that is independent of $d$. While Singhal and Steinke [2021] do not provide an answer to Question 1.1, they inspired our work to consider multiplicative singular-value gaps *in the input dataset* as a measure for easiness.

## 1.4 Defining Subspace Estimators

Towards answering Question 1.1, we consider mechanisms M that are parameterized by $k$, $\lambda$, and $\beta$, and satisfy the following utility guarantee: Given a dataset $X = (x_1, \ldots, x_n) \in (\mathcal{S}_d)^n$ and a value $\gamma$ as inputs, such that $X$ is "$\gamma$-easy" for $k$-subspace estimation, then with probability at least $\beta$ over a random execution of $\mathsf{M}(X, \gamma)$, the output $\Pi$ is an $\lambda\gamma$-useful rank-$k$ projection matrix for $X$.[2] In Definition 1.3, the "$\gamma$-easy" property is abstracted by a predicate $f$. We also allow an additional parameter $\gamma_{\max}$ to relax the utility for non-easy instances (i.e., we would not require a utility guarantee for instances that are not "$\gamma_{\max}$-easy"). Furthermore, we only focus on cases in which $X$ has at least $k$ significant directions, which is formalized by requiring that $\sigma_k^2(X) \geq 0.01 \cdot n/k$ (we refer to Remark 2.3 for how our upper bounds, stated in Theorem 1.6, can handle smaller values of $\sigma_k$ using an additional parameter or different assumptions).

**Definition 1.3** $((k, \lambda, \beta, \gamma_{\max}, f)$-Subspace Estimator)**.** *Let $n, k, d \in \mathbb{N}$ s.t. $k \leq \min\{n, d\}$, let $\beta \in (0, 1]$, and let $f \colon (\mathcal{S}_d)^n \times [0, 1] \to \{0, 1\}$ be a predicate. We say that $\mathsf{M} \colon (\mathcal{S}_d)^n \times [0, 1] \to \mathbb{R}^{d \times d}$ is an $(k, \lambda, \beta, \gamma_{\max}, f)$-subspace estimator, if for every $X \in (\mathcal{S}_d)^n$ and $\gamma \leq \gamma_{\max}$ with $\sigma_k^2(X) \geq 0.01 \cdot n/k$ and $f(X, \gamma) = 1$, it holds that*

$$\Pr_{\Pi \sim \mathsf{M}(X, \gamma)}[\Pi \text{ is } (\lambda\gamma)\text{-useful for } X] \geq \beta.$$

## 1.5 Quantifying Easiness - Our Approach

In this work, we develop two types of smooth measures (captured by the predicate $f$ in Definition 1.3) for input "easiness", which are translated to the following two types of subspace estimators:

---

[2]Similarly to the DP averaging example, we consider algorithms which guarantee accuracy that is proportional to the "easiness" parameter $\gamma$, and we measure the "quality" of the estimations by the parameter $\lambda$.

**Definition 1.4** (($k, \lambda, \beta, \gamma_{\max}$)-Weak Subspace Estimator). $\mathsf{M}\colon (\mathcal{S}_d)^n \times [0,1] \to \mathbb{R}^{d \times d}$ *is called an* ($k, \lambda, \beta, \gamma_{\max}$)-*Weak Subspace Estimator, if it is* ($k, \lambda, \beta, \gamma_{\max}, f$)-*Subspace Estimator for the predicate* $f(X, \gamma)$ *that outputs* 1 *iff* $\sum_{i=k+1}^{n} \sigma_i^2(X) \leq \gamma^2 \sigma_k^2(X)$.

**Definition 1.5** (($k, \lambda, \beta, \gamma_{\max}$)-Strong Subspace Estimator). $\mathsf{M}\colon (\mathcal{S}_d)^n \times [0,1] \to \mathbb{R}^{d \times d}$ *is called an* ($k, \lambda, \beta, \gamma_{\max}$)-*Strong Subspace Estimator, if it is* ($k, \lambda, \beta, \gamma_{\max}, f$)-*Subspace Estimator for the predicate* $f(X, \gamma)$ *that outputs* 1 *iff* $\sigma_{k+1}(X) \leq \gamma \sigma_k(X)$.

In both cases, we define $f$ based on multiplicative singular-value gaps in the input dataset, but the difference is what type of gap the value $\gamma$ bounds: Strong estimators depend solely on the gap $\sigma_{k+1}/\sigma_k$ without taking into account smaller singular values. Weak estimators, on the other, depend on the gap $\sqrt{\sum_{i=k+1}^{\min\{n,d\}} \sigma_i^2}/\sigma_k$. Note that a strong estimator is, in particular, a weak one (with the same parameters). Also note that both measures smoothly converge to the exact $k$-subspace case: When each gap tends to zero, the points tend to be closer to a $k$-dimensional subspace.

We provide new upper and lower bounds for both types of estimators.

### 1.5.1 Our Upper Bounds

**Theorem 1.6** (Weak estimator). *There exists an* ($k, \lambda, \beta = 0.9, \gamma_{\max} = \Theta(\min\{\frac{1}{\lambda}, 1\})$)-*weak subspace estimator* $\mathsf{M}\colon (\mathcal{S}_d)^n \times [0,1] \to \mathbb{R}^{d \times d}$ *with* $n = \tilde{O}_{\varepsilon, \delta}\left(k + \frac{\min\{k^2\sqrt{d}, kd\}}{\lambda}\right)$ *such that* $\mathsf{M}(\cdot, \gamma)$ *is* ($\varepsilon, \delta$)-*DP for every* $\gamma \in [0, 1]$.

**Theorem 1.7** (Strong estimator). *There exists an* ($k, \lambda, \beta = 0.8, \gamma_{\max} = \tilde{\Theta}(\min\{\frac{1}{\lambda}, \frac{\lambda^2}{k^2 d}\})$)-*strong subspace estimator* $\mathsf{M}\colon (\mathcal{S}_d)^n \times [0,1] \to \mathbb{R}^{d \times d}$ *with* $n = \widetilde{O}_{\varepsilon, \delta}\left(k + \frac{k^3 d}{\lambda^2}\right)$ *such that* $\mathsf{M}(\cdot, \gamma)$ *is* ($\varepsilon, \delta$)-*DP for every* $\gamma \in [0, 1]$.

Both of our estimators provide a useful projection by outputting a matrix that is close (in Frobenius norm) to the projection matrix of the top-$k$ rows subspace. Their running time is $\frac{n}{m} \cdot T(m, d, k) + \tilde{O}(dkn)$ for some $m = \tilde{\Theta}(k)$, where $T(m, d, k)$ denotes the running time required to compute (non-privately) a projection matrix to the top-$k$ rows subspace of an $m \times d$ matrix. We refer to Section 2 for a detailed overview.

For simplifying the presentation and the formal utility guarantees, we assume that our algorithms know the values of $\gamma$ (the bound on the multiplicative-gap) and of $k$ beforehand. Yet, we show that both assumptions are not inherent, and we refer to Remark 2.4 for additional details.

We also remark that in both theorems, it is possible to increase the confidence $\beta$ to any constant smaller than 1 without changing the asymptotic cost.

### 1.5.2 Our Lower Bounds

**Theorem 1.8** (Lower bound for weak estimators). *If* $\mathsf{M}\colon (\mathcal{S}_d)^n \times [0,1] \to \mathbb{R}^{d \times d}$ *is a* ($k, \lambda, \beta = 0.1, \gamma_{\max} = \Theta(\frac{1}{\lambda})$)-*weak subspace estimator for* $1 \leq \lambda \leq \Theta(\frac{d}{k \log k})$ *and* $\mathsf{M}(\cdot, \gamma)$ *is* $\left(1, \frac{1}{50nk}\right)$-*DP for every* $\gamma \in [0, 1]$, *then* $n \geq \tilde{\Omega}\left(\frac{\sqrt{k d}}{\lambda}\right)$.

**Theorem 1.9** (Lower bound for strong estimators). *If* $\mathsf{M}\colon (\mathcal{S}_d)^n \times [0,1] \to \mathbb{R}^{d \times d}$ *is a* ($k, \lambda, \beta = 0.1, \gamma_{\max} = \Theta(\frac{1}{\lambda})$)-*strong subspace estimator for* $1 \leq \lambda \leq \Theta(\frac{d}{\log k})$ *and* $\mathsf{M}(\cdot, \gamma)$ *is* $\left(1, \frac{1}{50nk}\right)$-*DP for every* $\gamma$, *then* $n \geq \tilde{\Omega}\left(\frac{k\sqrt{d}}{\lambda}\right)$.

Our lower bounds are more technically involved, and use a novel combination of generating hard-instances using the tools from Peter et al. [2024] for proving smooth lower bounds, and extracting sensitive vectors from useful projection matrices using ideas from the lower bound of Dwork et al. [2014]. Both lower bounds are proven by generating hard-instances that are "$\gamma$-easy" for $\gamma = \frac{1}{1000\lambda}$.

We refer to Section 3 for a detailed overview.

We remark that Peter et al. [2024] recently proved a similar lower bound for the special case of $k = 1$ (estimating the top-singular vector). However, their result strongly relies on the similarity between

|  | Weak Estimator | Strong Estimator |
|---|---|---|
| Upper Bound | $\tilde{O}_{\varepsilon,\delta}\left(k + \frac{k^2\sqrt{d}}{\lambda}\right)$ | $\widetilde{O}_{\varepsilon,\delta}\left(k + \frac{k^3 d}{\lambda^2}\right)$ |
| Lower Bound | $\tilde{\Omega}\left(\frac{\sqrt{kd}}{\lambda}\right)$ | $\tilde{\Omega}\left(\frac{k\sqrt{d}}{\lambda}\right)$ |

**Table 1:** Our bounds on $n$ for subspace estimation (ignoring restrictions on $\gamma_{\max}$ and $\lambda$).

|  | Weak Estimator | Strong Estimator |
|---|---|---|
| Upper Bound | $\tilde{O}_{\varepsilon,\delta}\left(k + \gamma_1 k^2\sqrt{d}\right)$ | $\widetilde{O}_{\varepsilon,\delta}\left(k + \gamma_2^2 k^3 d\right)$ |
| Lower Bound | $\tilde{\Omega}\left(\gamma_1\sqrt{kd}\right)$ | $\tilde{\Omega}\left(\gamma_2 k\sqrt{d}\right)$ |

**Table 2:** Our bounds on $n$ for computing 0.001-useful projection, where $\gamma_1 = \sqrt{\sum_{i=k+1}^{\min\{n,d\}} \sigma_i^2}/\sigma_k$ and $\gamma_2 = \sigma_{k+1}/\sigma_k$ denote here the "easiness" parameters for weak and strong estimators, respectively. As mentioned in Footnote 3, our upper bound for strong estimators only hold for $\gamma_2 \leq \Theta(k^{-2/3}d^{-1/3})$.

averaging and estimating top-singular vector in their hard instances, which does not hold for the case $k \geq 2$. Table 1 summarizes our bounds for $k \leq \sqrt{d}$.

### 1.5.3 Implications

We offer two formulations which have the property we seek: If we aim for an error $\alpha$, and the dataset is "$\gamma$-easy" for a very small $\gamma$, we take $\lambda = \alpha/\gamma$ to reduce the number of necessary and sufficient points.

For strong estimators, the rate $n = n(\lambda)$ in Theorem 1.7 does not match the corresponding lower bound Theorem 1.9, and Theorem 1.7 is limited to small values of $\gamma$.[3] Yet, for small values of $k$, the strong-estimator bounds do imply that in order to privately compute an 0.001-useful rank-$k$ projection with number of points that is independent of $d$, it is sufficient and necessary to have a gap $\sigma_{k+1}/\sigma_k$ of at most $\propto 1/\sqrt{d}$. Table 2 summarizes our bounds for the special case of outputing 0.001-useful projection matrix, using our two different types of input "easiness".

### 1.6 Empirical Evaluation

We believe that private subspace estimation of easy instances could find practical applications. Therefore, we made an effort to realize our upper bounds using a *practical* algorithm. In Section 4 we empirically compared our method to the additive-gap based approach of Dwork et al. [2014] for the task of privately estimating the empirical mean of points that are close to a small dimensional subspace, demonstrating the advantage of our approach in high-dimensional regimes.

### 1.7 Paper Organization

In Sections 2 and 3 we present proof overviews for our results, and the empirical evaluation is provided in Section 4. A limitations section is given in Section 5. Additional related work appears at Appendix A. Notations, definitions and general statements are given in Appendix B. Our upper bounds are proven in Appendix C, and our lower bounds are proven in Appendix D.

## 2 Upper Bounds - Overview

Both of our estimators (Theorems 1.6 and 1.7) follow the same structure, but with different parameters. Similarly to Singhal and Steinke [2021], our algorithms follow the sample-and-aggregate approach of Nissim et al. [2007]. That is, given a dataset $X = (x_1, \ldots, x_n) \in (\mathcal{S}_d)^n$, we partition the rows into $t$ subsets, compute (non-privately) a projection matrix to the top-$k$ rows subspace of each subset,

---

[3]If we take $\lambda = \alpha/\gamma$ (i.e., aiming for an error $\alpha$), the utility restriction on $\gamma_{\max}$ in Theorem 1.7 implies that $\gamma$ should be smaller than $\Theta\left(\frac{\alpha^{2/3}}{k^{2/3}d^{1/3}}\right)$.

and then privately aggregate the projection matrices $\Pi_1, \ldots, \Pi_t$. For doing that, we need to argue that $\Pi_1, \ldots, \Pi_t$ are expected to be close to each other. In the Gaussian setting of Singhal and Steinke [2021], this holds by concentration properties of Gaussian distributions. In our setting, however, it is unreasonable to expect that arbitrary partitions will lead to similar subspaces. For instance, consider the matrix $X$ whose first $n/k$ rows are $e_1 = (1, 0, \ldots, 0)$, the next $n/k$ rows are $e_2 = (0, 1, \ldots, 0)$, and so forth until $e_k$. Even though $X$ is rank-$k$ and has $\sigma_1^2 = \ldots = \sigma_k^2 = n/k$, if we simply partition the rows according to their order, then most of those subsets will induce a rank-1 matrix which clearly does not represent the original matrix $X$. Therefore, we must consider a more clever partition that will guarantee a good representation of the top-$k$ rows subspace of $X$ in each subset.

There is an extensive line of works who aim for methods to choose a small subset of rows that provides a good low-approximation for the original matrix (e.g., see Mahoney [2011] for a survey). Yet, most of these methods are data-dependent, and therefore seem less applicable for privacy.

In this work we show that by simply using a *uniformly random* partition into subsets of size $\tilde{\Theta}(k)$, then w.h.p. each subset induces a projection matrix that is close to the projection matrix of the top-$k$ rows subspace of $X$:

**Lemma 2.1.** *Let* $X = (x_1, \ldots, x_n) \in (\mathcal{S}_d)^n$ *with singular values* $\sigma_1 \geq \ldots \geq \sigma_{\min\{n,d\}} \geq 0$, *and let* $\gamma_1 = \frac{\sigma_{k+1}}{\sigma_k}$ *and* $\gamma_2 = \frac{\sqrt{\sum_{i=k+1}^{\min\{n,d\}} \sigma_i^2}}{\sigma_k}$. *Let* $\mathbf{X}' \in (\mathcal{S}_d)^m$ *be a uniformly random $m$-size subset of the rows of $X$ (without replacement). Let* $\Pi$ *and* $\mathbf{\Pi}'$ *be the projection matrices to the top-$k$ rows subspace of $X$ and $\mathbf{X}'$, respectively. Assuming that* $\sigma_k^2 \geq 0.01n/k$, *then the following holds for* $m = \tilde{\Theta}(k)$:

1. *If* $\gamma_1 \leq \frac{m}{2n}$, *then* $\Pr\left[\|\Pi - \mathbf{\Pi}'\| \leq O\left(\sqrt{\frac{n}{m}} \cdot \gamma_1\right)\right] \geq 0.9$. *($\|\cdot\|$ denotes the Spectral norm[4]).*

2. *If* $\gamma_2 \leq 0.1$, *then* $\Pr[\|\Pi - \mathbf{\Pi}'\|_F \leq O(\gamma_2)] \geq 0.9$.[5]

Namely, Item 1 bounds the expected spectral norm distance of the projection matrices using the first type gap $\sigma_{k+1}/\sigma_k$ (which is used in the analysis of our strong estimator), and Item 2 bounds the expected Frobenius norm distance using the second type gap $\sqrt{\sum_{i=k+1}^{\min\{n,d\}} \sigma_i^2}/\sigma_k$ (which is used in the analysis of our weak estimator). We prove Lemma 2.1 in Appendix C.1.2.

The next step is to aggregate the non-private projection matrices $\Pi_1, \ldots, \Pi_t$ into a private one $\widetilde{\Pi}$. We consider two types of aggregations. The first one simply treats each matrix as a $d^2$ vector and privately estimate the average of $\Pi_1, \ldots, \Pi_t$. The second type (which outperforms the first one in most cases) follows a similar high-level structure of Singhal and Steinke [2021]. That is, to sample i.i.d. reference points $p_1, \ldots, p_q \sim \mathcal{N}(\vec{0}, \mathbb{I}_{d \times d})$ for $q = \Theta(k)$, privately average the $qd$-dimensional points $\{(\Pi_j p_1, \ldots, \Pi_j p_q)\}_{j=1}^t$ for obtaining a private $\widetilde{P} \in \mathbb{R}^{q \times d}$ (whose $i^{\text{th}}$ row estimates the projection of $p_i$ onto the top-$k$ rows subspace of $X$), and then compute the projection matrix of the top-$k$ rows subspace of $\widetilde{P}$. But unlike Singhal and Steinke [2021] who perform this step using a histogram-based averaging that has the same flavor of Karwa and Vadhan [2018], we perform this step using FriendlyCore Tsfadia et al. [2022] that simplifies the construction and makes it practical in high dimensional regimes. We remark that in both aggregation types, we need a DP averaging algorithm that is resilient to a constant fraction of outliers (say, $20\%$) since both items in Lemma 2.1 only guarantee that the expected number of outliers is no more than $10\%$. Fortunately, FriendlyCore can be utilized for such regimes of outliers (see Appendix B.8.3 for more details).

A few remarks are in order.

**Remark 2.2.** *The first aggregation type (which privately estimate the average of $\Pi_1, \ldots, \Pi_t$ directly) outperforms the second type only for our weak estimator in the regime $k \geq \sqrt{d}$ (as it inherently posses larger dependency in the dimension).*

**Remark 2.3.** *We could avoid the requirement $\sigma_k^2 \geq 0.01n/k$ by adding an additional parameter $\eta$ such that $\sigma_k^2 \geq \eta \cdot n/k$, and using subsets of size $m = \tilde{\Theta}(k/\eta)$ (which would increase $n$ by the same factor of $1/\eta$). For readability purposes, we chose to avoid this additional parameter. Because our algorithms provide useful projection by estimating the actual top-$k$ projection, then such a*

---

[4]The spectral norm of a matrix $A \in \mathbb{R}^{n \times d}$ is defined by $\|A\| = \sup_{x \in \mathcal{S}_d} \|Ax\|_2$ and is equal to $\sigma_1(A)$.

[5]The Frobenius norm of a matrix $A \in \mathbb{R}^{n \times d}$ is equal to $\sqrt{\sigma_1(A)^2 + \ldots + \sigma_{\min\{n,d\}}(A)^2}$.

*requirement is unavoidable if we would like to provide utility guarantees only as a function of the singular values.[6] In fact, any assumption that would imply that two random subsets induce similar top-$k$ projection matrices would suffice for the utility analysis.*

**Remark 2.4.** *To eliminate the known-$\gamma$ assumption, we can replace the FriendlyCore-averaging step Tsfadia et al. [2022] (that requires to know the diameter) with their* unknown *diameter variant that gets two very rough bounds $\xi_{\min}$ and $\xi_{\max}$, and performs a private binary search for estimating a good diameter $\xi \in [\xi_{\min}, \xi_{\max}]$ in a preprocessing phase. This step only replaces the dependency on $d$ with $d + \log \log(\xi_{\max}/\xi_{\min})$ in the asymptotic sample complexity (section 5.1.2. in Tsfadia et al. [2022]) and is very practical. In fact, we use this method in our empirical evaluation in Section 4.*

*For handling unknown values of $k$, note that our algorithms provide useful utility guarantees (compared to the additive-gap based ones) only when $\sigma_{k+1}^2 \ll 1$. So in cases where $\sigma_k^2 \geq 2 \cdot \log(k/\beta)/\varepsilon'$ for some (fixed) privacy budget $\varepsilon' < \varepsilon$ (say, $\varepsilon' = \varepsilon/10$), we can privately determine w.p. $1 - \beta$ the right value of $k$ using a simple $\varepsilon'$-DP method. The main observation is that the $\ell_1$ sensitivity of the vector $(\sigma_1^2, \ldots, \sigma_n^2)$ is at most 2 (Amin et al. [2019]), yielding that we can privately compute $(\sigma_1'^2, \ldots, \sigma_n'^2) = (\sigma_1^2 + \mathrm{Lap}(2/\varepsilon'), \ldots, \sigma_n^2 + \mathrm{Lap}(2/\varepsilon'))$, and then perform analysis on $(\sigma_1'^2, \ldots, \sigma_n'^2)$ to set $k$ as the first index $i$ where $\sigma_i^2 \geq \log(i/\beta)/\varepsilon'$ and $\sigma_{i+1}^2 < \log(i/\beta)/\varepsilon'$.*

## 3 Lower Bounds - Overview

Our lower bounds (Theorems 1.8 and 1.9) use the recent framework of Peter et al. [2024] for generating smooth lower bounds for DP algorithms using Fingerprinting Codes (FPC), but require technically involved analysis due to the complex structure of this problem for $k \geq 2$.

Roughly speaking, let $\mathcal{D}$ be a distribution over $\{-1,1\}^{n_0 \times d_0}$ that induces an optimal FPC codebook with $d_0 = \tilde{O}(n_0^2)$ (e.g., Tardos [2008], Peter et al. [2024]). The connection between FPC and DP (first introduced by Bun et al. [2014]) is that any DP algorithm, given a random codebook $X = (x_i^j)_{i \in [n_0], j \in [d_0]} \sim \mathcal{D}$ as input, cannot output a vector $q = (q^1, \ldots, q^{d_0}) \in \{-1,1\}^{d_0}$ that "agrees" with most of the "marked" columns of $X$ (Formally, for $b \in \{-1,1\}$, a columns $x^j = (x_1^j, \ldots, x_n^j)$ is called $b$-marked if $x_1^j = \ldots = x_n^j = b$, and $q$ agrees with it if $q^j = b$).

Now consider a DP mechanism $\mathsf{M} \colon \mathcal{X}^n \to \mathcal{W}$ that satisfies some non-trivial accuracy guarantee. Peter et al. [2024] reduces the task of lower bounding $n$ to the following task: (1) Generate from an FPC codebook $X \in \{-1,1\}^{n_0 \times d_0}$ hard instances $Y \in \mathcal{X}^n$ for $\mathsf{M}$, and (2) Extract from the output $w \sim \mathsf{M}(Y)$ a vector $q \in \{-1,1\}^{d_0}$ that agrees with most of the marked columns of $X$ ($n_0$ and $d_0$ are some functions of $n$, $\mathcal{X}$ and the weak accuracy guarantee of $\mathsf{M}$). Peter et al. [2024] proved that if there exists such generating algorithm $\mathsf{G}$ and extracting algorithm $\mathsf{F}$ (which even share a random secret that $\mathsf{M}$ does not see) such that $\mathsf{G}$ is *neighboring-preserving* (i.e., maps neighboring databases to neighboring databases), then it must hold that $n_0 \geq \tilde{\Omega}(\sqrt{d_0})$ (Otherwise, $\mathsf{M}$ cannot be DP).

**Warm-up: DP averaging.** We first sketch how Peter et al. [2024] applied their framework with $n_0 = n$ and $d_0 = \Theta(d/\lambda^2)$ for proving a lower bound for the simpler problem of DP averaging. In this setting, we are given a mechanism that guarantees $\lambda\gamma$-accuracy ($\ell_2$ additive error) for $\gamma$-easy instances (i.e., points that are $\gamma$-close to each other in $\ell_2$ norm). The generator $\mathsf{G}$, given an FPC codebook $X \in \{-1,1\}^{n_0 \times d_0}$, uses the *padding-and-permuting* technique: It pads $\ell \approx 10^4 \lambda^2 d_0$ 1-marked columns and $\ell$ $(-1)$-marked columns, and then permutes all the $d = d_0 + 2\ell$ columns of the new codebook $X'$ using a random permutation $\pi \colon [d] \to [d]$ that is shared with the extractor $\mathsf{F}$. The input $Y$ to the algorithm would be the *normalized* rows of $X'$ which are $\frac{1}{100\lambda}$-close to each other in $\ell_2$ norm, so the mechanism has to output an $\frac{1}{100}$-accurate solution $w$. In particular, after rounding $w$ to $\{-1,1\}^d$, the coordinates of $w$ must agree with a vast majority of the marked columns, and also with a vast majority of the original marked columns that are located within $\pi(1), \ldots, \pi(d_0)$ as it cannot distinguish between them and the other marked columns (because $\pi$ is hidden from it). The extractor $\mathsf{F}$, given $w$ and $\pi$, rounds $w$ to $\{-1,1\}^d$ and outputs $q = (w^{\pi(1)}, \ldots, w^{\pi(d)})$ which agrees with most of the marked columns of $X$. Hence, we obtain the lower bound of $n \geq \tilde{\Omega}(\sqrt{d_0}) = \tilde{\Omega}(\sqrt{d}/\lambda)$.

---

[6]To illustrate why $\sigma_1, \ldots, \sigma_k$ should be large, consider a matrix $X$ whose first $n - k + 1$ rows are $e_1$, and the next $k - 1$ rows are $e_2, \ldots, e_k$. This matrix has $\sigma_2 = \ldots = \sigma_k = 1$, and even though it is a rank-$k$ matrix, it is clearly impossible to output a projection matrix that reveals any of the directions $e_2, \ldots, e_k$ under DP.

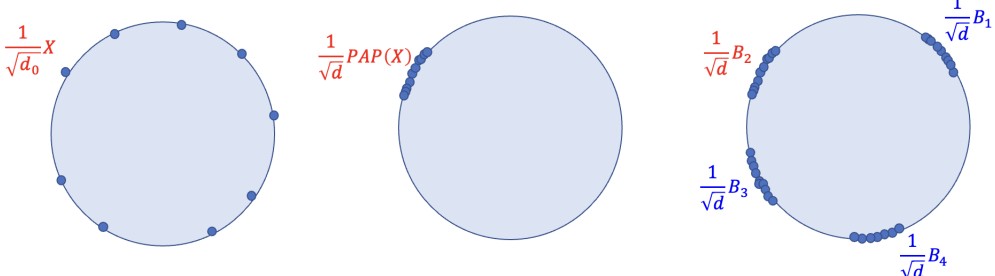

**Figure 1:** From Left to Right: (1) The normalized rows of the fingerprinting codebook $X$ are well-spread on the $d_0$-dimensional unit sphere. (2) Applying the padding-and-permuting (PAP) technique makes the normalized points very close to each other on the $d$-dimensional unit sphere ($d \gg d_0$). (3) We create hard instances for DP subspace estimation using $k$-independent (normalized) PAP-FPC codebooks $B_1, \ldots, B_k$, where $PAP(X)$ is planted in one of the $B_i$'s (in this example, in $B_2$). Reducing $\alpha$ (i.e., increasing the padding length) makes the points in each group $B_i$ closer to each other, which in particular, increases the closeness to a $k$-dimensional subspace.

**DP Subspace Estimation**  In our case, we are given a (weak or strong) subspace estimator $\mathsf{M} \colon (\mathcal{S}_d)^n \to \mathbb{R}^{d \times d}$ that outputs an $\lambda\gamma$-useful rank-$k$ projection matrix if $\sigma_{k+1}(X) \leq \gamma\sigma_k(X)$ (or $\sqrt{\sum_{i=k+1}^{\min\{n,d\}} \sigma_i(X)^2} \leq \gamma\sigma_k(X)$). We prove our lower bounds by applying the framework with $n_0 = n/k$ and $d_0 = \Theta(\alpha d)$, for some parameter $\alpha = \alpha(\lambda)$ that will depend on the type of $\mathsf{M}$ we consider. In order to generate hard instances $Y \in (\mathcal{S}_d)^n$ for $\mathsf{M}$ given an FPC codebook $X \sim \mathcal{D}$ ($X \in \{-1, 1\}^{(n/k) \times d_0}$), we use a variation of the approach to Dwork et al. [2014]. Namely, our generator $\mathsf{G}$ samples $k$ independent FPC codebooks $A_1, \ldots, A_k \sim \mathcal{D}$ where it plants $A_i = X$ for a random $i \leftarrow [k]$. Then for each $j \in [k]$, it applies (independently) the padding-and-permuting technique of Peter et al. [2024] where it pads $\ell$ 1-marked columns and $\ell$ ($-1$)-marked columns for $\ell \approx \frac{d_0}{2\alpha}$, and permute all the columns. This induces $k$ matrices $B_1, \ldots, B_k \in \{-1, 1\}^{(n/k) \times d}$ (for $d = d_0 + 2\ell$) such that each $B_j$ is "almost" rank-1 and their vertical concatenation $B \in \{-1, 1\}^{n \times d}$ is almost rank-$k$. It provides $Y = \frac{1}{\sqrt{d}}B$ as the input for $\mathsf{M}$. See Figure 1 for graphical illustrations.

We remark that at this step, the main difference from Dwork et al. [2014] (who implicitly follow a similar paradigm) is that they use a fixed padding length of $\ell = 15d_0$ that suffices for the robustness properties that they need. On the other hand, we use Peter et al. [2024]'s observation that increasing the padding can handle low-accuracy regimes of many problems, and indeed we use the padding length $\ell$ to increase the $k$'th singular value gap, which will be a function of the quality parameter $\lambda$.

The next step is to choose the right value of $\alpha = \alpha(\lambda)$ such that $\mathsf{M}$, on input $Y$, will have to output a useful projection matrix. We show that the input matrix $Y$ has w.h.p. $\sigma_1(Y)^2 \geq \ldots \geq \sigma_k(Y)^2 \geq (1 - O(\alpha)) \cdot \frac{n}{k}$, which yields that $\sum_{i=k+1}^{\min\{n,d\}} \sigma_i(Y)^2 \leq O(\alpha)n$. If $\mathsf{M}$ is a weak estimator, then we simply use $\alpha = \Theta\left(\frac{1}{\lambda^2 k}\right)$ to guarantee that $\sum_{i=k+1}^{\min\{n,d\}} \sigma_i(Y)^2 \leq \gamma^2 \cdot \sigma_k(Y)^2$ for $\gamma = \frac{1}{1000\lambda}$, which yields that by the utility guarantee of $\mathsf{M}$, we get an 0.001-useful projection matrix. If $\mathsf{M}$ is a strong estimator, then we use $\alpha = \Theta\left(\frac{1}{\lambda^2}\right)$ (i.e., we decrease the padding length by a factor of $k$). Yet, in order to meet the requirements of $\mathsf{M}$, we must argue that w.h.p., $\sigma_{k+1}(Y)^2 \leq \tilde{O}(\alpha) \cdot \frac{n}{k}$, and this is more complex than the previous case. Here we use more internal properties of the fingerprinting distribution $\mathcal{D}$. Namely, that in Peter et al. [2024]'s construction (which is also true for Tardos [2008]'s one), each entry of the codebook matrix has expectation 0 and the columns of the matrix are independent. Using known concentration bounds, this allows us to argue that if we pick a unit vector $v$ that is orthogonal to the top-$k$ rows subspace of $Y$, then with probability at least $1 - \exp(-\tilde{\Omega}(d))$ it holds that $\|Yv\|_2^2 \leq \tilde{O}(\alpha) \cdot \frac{n}{k}$. Since $\sigma_k^2$ is bounded by the supremum of $\|Yv\|_2^2$ under such unit vectors, we conclude the proof of this part using a net argument.

Finally, the last step, which is not trivial for $k \geq 2$, is to extract from an 0.001-useful projection matrix $\widetilde{\Pi}$ for $Y$, a vector $q \in \{-1, 1\}^{d_0}$ that with noticeable probability, strongly agrees with the marked columns of the original codebook $X \in \{-1, 1\}^{n_0 \times d_0}$. For that, our extractor $\mathsf{F}$ uses the random permutations and the random location $i$ (which are part of the shared secret between the

generator and the extractor) and follows the strategy of Dwork et al. [2014]. That is, it applies the $i^{\text{th}}$ invert permutation over the columns of $\widetilde{\Pi}$ (denote the resulting matrix by $\widetilde{\Pi}_i$), chooses a vector $u \in Span(\widetilde{\Pi}_i)$ that has the maximal agreement with *half* of the padding bits, and then simply outputs its first $d_0$ coordinates after rounding to $\{-1, 1\}$. The intuition is that an 0.001-useful projection matrix must be also 0.001-useful for at least one of the parts $Y_j = \frac{1}{\sqrt{d}} B_j$. Since all the $Y_j$'s have the same distribution and the location $i$ (where the original $X$ is planted) is hidden from M, then it must be 0.001-useful for $Y_i$ with probability at least $\beta/k$ (where $\beta$ denotes the success probability of M). Given that this event occurs, the usefulness of $\widetilde{\Pi}$ implies that there must exists a vector in $Span(\widetilde{\Pi}_i)$ that strongly agrees with half of the padding locations. But because all the marked columns (that includes the padding locations) are indistinguishable from the eyes of M who computed $\widetilde{\Pi}$, then a similar agreement must hold for the marked columns of $X$.

## 4 Empirical Evaluation

We implemented a zCDP (Definition B.24) variant of our subspace estimation algorithm in Python (denoted by EstSubspace), and in this section we present empirical results for the fundamental task of privately estimating the average of $d$ dimensional points that approximately lie in a much smaller $k$-dimensional subspace. Namely, given a dataset $X = (x_1, \ldots, x_n) \in \mathcal{S}_d^n$, a parameter $k$, and zCDP parameters $\rho, \delta$, we perform the following steps: (a) Compute a $(\rho/2, \delta)$-zCDP rank-$k$ projection matrix $\widetilde{\Pi}$ using EstSubspace that estimates the projection onto the top-$k$ rows subspace of $X$, (b) Compute a $\rho/2$-zCDP estimation of the average of $X$ using the Gaussian Mechanism: $\tilde{x} = \frac{1}{n} \sum_{i=1}^{n} x_i + \mathcal{N}(\mathbf{0}, \sigma^2 \cdot \mathbb{I}_{d \times d})$ for $\sigma = \frac{2}{n\sqrt{\rho}}$, and (c) Output $\hat{x} = \widetilde{\Pi} \cdot \tilde{x}$.

The accuracy is measured by the $\ell_2$ error from the average: $\left\| \hat{x} - \frac{1}{n} \sum_{i=1}^{n} x_i \right\|_2$.

In all our experiments, we use $\rho = 2$ and $\delta = 10^{-5}$, $t = 125$ (the number of subsets in the sample-and-aggregate process), $n = 2tk$ data points, $q = 10 \cdot k$ (the number of reference points in the aggregation), and use the zCDP implementation of the FriendlyCore-based averaging algorithm of Tsfadia et al. [2022].[7] All experiments were tested on a MacBook Pro Laptop with 8-core Apple M1 CPU with 16GB RAM.

Rather than using Tsfadia et al. [2022]'s algorithm for the known-diameter case, we use their unknown-diameter implementation with $\xi_{\min} = 10^{-6}$ and $\xi_{\max} = 100$ (see Remark 2.4 for details). Furthermore, we reduced the space complexity of our implementation from $\tilde{\Theta}(d^2)$ to $\tilde{\Theta}(kd)$.[8]

In order to generate a synthetic dataset that approximately lie in a $k$-dimensional subspace, we initially sample uniformly random $b_1, \ldots, b_k \leftarrow \{-1, 1\}^d$ and perform the following process to generate each data point: (i) Sample a random unit vector $u$ in $Span\{b_1, \ldots, b_k\}$, (ii) Sample a random noise vector $\nu \leftarrow \{1/\tau, -1/\tau\}^d$, and (iii) Output $\frac{u+\nu}{\|u+\nu\|}$ (note that higher $\tau$ results with data points that are closer to a $k$-dimensional subspace).

We compare our averaging method to two other approaches: The first one simply applies the Gaussian mechanism directly on $X = (x_1, \ldots, x_n)$ using the entire privacy budget $\rho$ (i.e., without computing a projection matrix). The second one replaces our Step (a) by computing the projection matrix $\widetilde{\Pi}$ using a $(\rho/2, \delta)$-zCDP variant of the additive-gap based algorithm of Dwork et al. [2014] (see Appendix B.8.4 for more details). [9] The empirical results are presented in Figure 2. In all experiments, we perform 30 repetitions for generating each graph point which represents the trimmed average of

---

[7]Their source code is publicly available in `https://media.icml.cc/Conferences/ICML2022/supplementary/tsfadia22a-supp.zip`.

[8]We do not explicitly compute a $d \times d$ rank-$k$ projection matrix in each subset, but rather only compute a good approximation of the top-$k$ rows $V = (v_1, \ldots, v_k) \in \mathcal{S}_d^k$ using the Python function randomized_svd (provided in the sklearn library). We then compute the projection of any vector $u \in \mathbb{R}^d$ onto $Span\{v_1, \ldots, v_k\}$, given by $V^T V u$, from right to left, which only involves $O(kd)$ time and space computation cost. We do the same thing w.r.t. to the output projection $\widetilde{\Pi}$ (i.e., represent it using only $k$ vectors).

[9]We remark that unlike EstSubspace, Dwork et al. [2014]'s algorithm requires $O(d^2)$ time and space complexity as it requires an explicit access to the $d \times d$ projection matrix onto the top-$k$ rows subspace, and therefore is limited to moderate values of $d$. Still, we were able to use it as baseline since we saw the advantage of our approach in terms of accuracy even when $d$ is not extremely high.

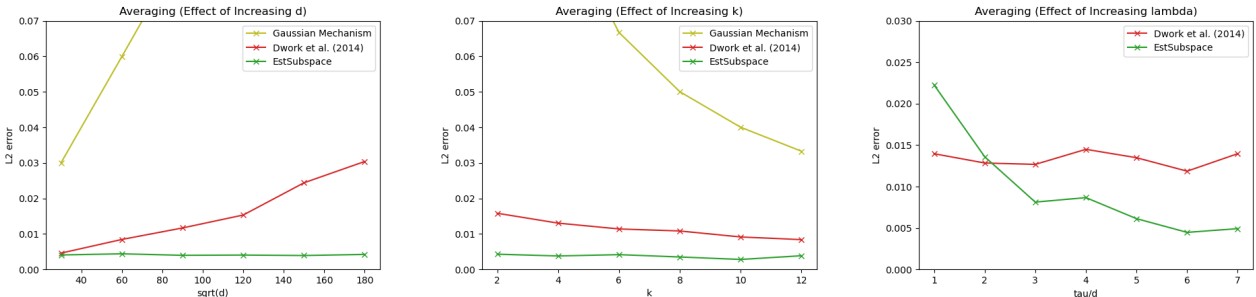

**Figure 2:** From Left to Right: (1) The case $k = 4$ and $\tau = 10d$, varying $d$ (the $X$-axis is $\sqrt{d}$). (2) The case $d = 10^4$ and $\tau = 10d$, varying $k$. (3) The case $d = 10^4$ and $k = 4$, varying $\tau$ (the $X$-axis is $\tau/d$). In all the experiments, we use $n = 250 \cdot k$ data points.

values between the $0.1$ and $0.9$ quantiles. We show the $\ell_2$ error of our estimate on the $Y$-axis. The first graph illustrates the inherent dependency on $d$ that Dwork et al. [2014]'s algorithm has, while our algorithm EstSubspace takes advantage of the closeness of the points to dimension $k$ in order to eliminate this dependency. The second graph illustrates that when $d$ is fixed, increasing $k$ and $n$ in the same rate a has similar affect on both EstSubspace and Dwork et al. [2014]'s algorithm. In the last graph we compare the accuracy of EstSubspace and Dwork et al. [2014]'s algorithm as a function of the closeness to a subspace $k$ (measured in our experiments by the parameter $\tau$), and show in what regimes EstSubspace outperforms Dwork et al. [2014]'s algorithm.

## 5 Limitations and Future Directions

From a theoretical perspective, our work is the first to provide proper measures for how "easy" a given dataset is which smoothly eliminates the dependency on the dimension $d$. Yet, closing the gap between our upper and lower bounds is still left open. Specifically, for weak estimator, there is still a gap of $k^{1.5}$ between Theorems 1.6 and 1.8. For strong estimators, the upper-bound rate $n = n(\lambda)$ (Theorem 1.7) does not align with the one of the lower bound (Theorem 1.9), and it is left open to relax the restriction on $\gamma_{\max}$. One possible reason for some of these gaps (especially the dependency in $k$) is that our upper bounds follow the approach of Singhal and Steinke [2021] to estimate (under some matrix norm) the projection matrix to the top-$k$ rows subspace (we do it in Frobenius norm). While estimating the projection matrix itself provides, in particular, a useful solution (Proposition B.16), the opposite direction is not true in general, and it could be possible to reduce the sample complexity by focusing on $\alpha$-usefulness (Definition 1.2) directly, or alternatively, providing stronger lower bounds for estimating the projection matrix.

From a more practical standpoint, we empirically demonstrate the advantage of our approach in high-dimension regimes when the data is very close to a low-dimensional structure, which is directly translated to an advantage in private mean estimation of such instances. The downside of our approach is that it requires the points to be very close to a $k$-dimensional structure in order to be effective, which might not be sufficient for typical training scenarios in deep learning. It would be intriguing to explore if there is a connection between training parameters (e.g., the network structure) to the phenomena of gradients that are close to a low-dimensional subspace (mentioned in Section 1.1). If we could boost this closeness to regimes where our method achieves high accuracy, we could generate drastically improved private models. On the other hand, if we cannot do it, then our lower bounds indicate that improving DP-SGD via private subspace estimation might not be the right approach, and we should focus on different approaches for this task.

## Acknowledgments and Disclosure of Funding

Work supported by a gift to Georgetown University.

The author would like to thank Edith Cohen and Jonathan Ullman for very useful discussions.

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

# A  Additional Related Work

A closely related line of work is on Private PCA. Dwork et al. [2014] consider the simple algorithm that adds independent Gaussian noise to each entry of the covariance matrix $A = \sum_{i=1}^{n} x_i^T x_i \in \mathbb{R}^{d \times d}$, and then performs analysis on the noisy matrix. This method, predating the development of differential privacy Blum et al. [2005], was later analyzed under differential privacy by McSherry and Mironov [2009] and Chaudhuri et al. [2013]. This simple algorithm is versatile and several bounds are provided for the accuracy of the noisy PCA. The downside of this is that a polynomial dependence on the ambient dimension $d$ is inherent for any instances (including "easy" ones). While this approach has a variant that improves the accuracy of estimating the top-$k$ subspace as a function of the additive gap $\sigma_k^2 - \sigma_{k+1}^2$ (Appendix B.8.4), it does not prevent the polynomial dependency on the dimension $d$ even for very easy instances.

Techniques from dimensionality reduction have been applied by Hardt and Roth [2012] and Arora et al. [2018] to privately compute a low-rank approximation to the input matrix $X$. Similarly, Hardt and Roth [2013] and Hardt and Price [2014] utilize the power iteration method with noise injected at each step to compute low-rank approximations to $X$. Despite their effectiveness, these methods, relying on noise addition, require sample complexity to grow polynomially with the ambient dimension to achieve meaningful guarantees.

Another approach, employed by Blocki et al. [2012] and Sheffet [2019], involves approximating the covariance matrix $A$ using dimensionality reduction. They show that the dimensionality reduction step itself provides a privacy guarantee (whereas the aforementioned results did not exploit this and relied on noise added at a later stage).

Chaudhuri et al. [2012], Kapralov and Talwar [2013], Wei et al. [2016] apply variants of the exponential mechanism McSherry and Talwar [2007] to privately select a low-rank approximation to the covariance matrix $A$. This method is nontrivial to implement and analyse, but it ultimately requires the sample complexity to grow polynomially in the ambient dimension.

Gonem and Gilad-Bachrach [2018] exploit smooth sensitivity Nissim et al. [2007] to release a lowrank approximation to the matrix $A$. This allows adding less noise than worst case sensitivity, under an eigenvalue gap assumption. However, the sample complexity $n$ is polynomial in the dimension $d$.

Another related area involves estimating the parameters of unbounded Gaussians Kamath et al. [2022], Ashtiani and Liaw [2022], Kothari et al. [2022], Tsfadia et al. [2022]. Notably, Kamath et al. [2022] used the subspace learning algorithm of Singhal and Steinke [2021] to efficiently learn the covariance matrix.

A recent popular trend in DP learning is to utilize a few *public* examples to enhance accuracy. This has led to methods for private ML which project the sensitive gradients onto a subspace estimated from the public gradients. By using a small amount of i.i.d. public data, Zhou et al. [2021] demonstrate that this approach can improve the accuracy of differentially private stochastic gradient descent in high-privacy regimes and achieve a dimension-independent error rate. Similarly, Yu et al. [2021] proposed GEP, a method that utilizes public data to identify the most useful information carried by gradients, and then splits and clips them separately. These works underscore the importance of identifying the subspace of gradients in private ML.

# B  Preliminaries

## B.1  Notations

We use calligraphic letters to denote sets and distributions, uppercase for matrices and datasets, boldface for random variables, and lowercase for vectors, values and functions. For $n \in \mathbb{N}$, let $[n] = \{1, 2, \ldots, n\}$. Throughout this paper, we use $i \in [n]$ as a row index, and $j \in [d]$ as a column index (unless otherwise mentioned).

For a matrix $X = (x_i^j)_{i \in [n], j \in [d]}$, we denote by $x_i$ the $i^{\text{th}}$ row of $X$ and by $x^j$ the $j^{\text{th}}$ column of $X$. A column vector $x \in \mathbb{R}^n$ is written as $(x_1, \ldots, x_n)$ or $x = x_{1 \ldots n}$, and a row vector $y \in \mathbb{R}^d$ is written as $(y^1, \ldots, y^d)$ or $y^{1 \ldots d}$. In this work we consider mechanisms who receive an $n \times d$ matrix $X$ as input, which is treated as the dataset $X = (x_1, \ldots, x_n)$ where the rows of $X$ are the elements (and therefore, we sometimes write $X \in (\mathbb{R}^d)^n$ instead of $X \in \mathbb{R}^{n \times d}$ to emphasize it).

For a vector $x \in \mathbb{R}^d$ we define $\|x\|_2 = \sqrt{\sum_{i=1}^n x_i^2}$ (the $\ell_2$ norm of $x$), and for a subset $\mathcal{S} \subseteq [d]$ we define $x_{\mathcal{S}} = (x_i)_{i \in \mathcal{S}}$, and in case $x$ is a row vector we write $x^{\mathcal{S}}$. Given two vectors $x = (x_1, \ldots, x_n), y = (y_1, \ldots, y_n)$, we define $\langle x, y \rangle = \sum_{i=1}^n x_i y_i$ (the inner-product of $x$ and $y$). We denote by $\mathcal{S}_d$ the set of $d$-dimensional unit vectors, that is, $\mathcal{S}_d = \{v \in \mathbb{R}^d \colon \|v\|_2 = 1\}$. For a matrix $X = (x_i^j)_{i \in [n], j \in [d]}$ we let $\|X\| = \max_{v \in \mathcal{S}_d} \|Xv\|_2$ (the *spectral norm* of $X$) and let $\|X\|_F = \sqrt{\sum_{i \in [n], j \in [d]} (x_i^j)^2}$ (the Forbenius norm of $X$). For a matrix $X = (x_i^j)_{i \in [n], j \in [d]} \in \{-1, 1\}^{n \times d}$ and $b \in \{-1, 1\}$, we define the $b$-marked columns of $X$ as the subset $\mathcal{J}_X^b \subseteq [d]$ defined by $\mathcal{J}_X^b = \{j \in [d] \colon x_i^j = b \text{ for all } i \in [n]\}$.

For $d \in \mathbb{N}$ we denote by $\mathcal{P}_d$ the set of all $d \times d$ permutation matrices. For a permutation matrix $P \in \mathcal{P}_d$ and $i \in [d]$ we denote by $P(i)$ the index $j \in [d]$ for which $e_i P = e_j$ (where $e_i$ and $e_j$ are the corresponding one-hot row vectors), and for $\mathcal{I} \subseteq [d]$ we denote $P(\mathcal{I}) = \{P(i) \colon i \in \mathcal{I}\}$.

For $d, k \in \mathbb{N}$ we denote by $\mathcal{W}_{d,k}$ the set of all $d \times d$ projection matrices of rank $k$. For a matrix $A \in \mathbb{R}^{n \times d}$ we denote by $Span(A)$ the columns subspace of $A$ (and therefore the rows subspace of $A$ is $Span(A^T)$).

For $z \in \mathbb{R}$, we define $\mathsf{sign}(z) := \begin{cases} 1 & z \geq 0 \\ -1 & z < 0 \end{cases}$ and for $v = (v^1, \ldots, v^d) \in \mathbb{R}^d$ we define $\mathsf{sign}(v) := (\mathsf{sign}(v^1), \ldots, \mathsf{sign}(v^d)) \in \{-1, 1\}^d$.

## B.2   Distributions and Random Variables

Given a distribution $\mathcal{D}$, we write $x \sim \mathcal{D}$ to denote that $x$ is sampled according to $\mathcal{D}$. For a set $\mathcal{S}$, we write $x \leftarrow \mathcal{S}$ to denote that $x$ is sampled from the uniform distribution over $\mathcal{S}$.

## B.3   Singular Value Decomposition (SVD)

For a matrix $X \in \mathbb{R}^{n \times d}$, the singular value decomposition of $X$ is defined by $X = U\Sigma V^T$, where $U \in \mathbb{R}^{n \times n}$ and $V \in \mathbb{R}^{d \times d}$ are unitary matrices. The matrix $\Sigma \in \mathbb{R}^{n \times d}$ is a diagonal matrix with non-negative entries $\sigma_1 \geq \ldots \geq \sigma_{\min\{n,d\}} \geq 0$ along the diagonal, called the singular values of $X$. The SVD of $X$ can also be written in the form $X = \sum_{i=1}^{\min\{n,d\}} \sigma_i u_i v_i^T$, where $u_i$ and $v_i$ are the $i^{\text{th}}$ columns of $U$ and $V$ (respectively). It holds that $\|X\|_F^2 = \sum_i \sigma_i^2$ and $\|X\| = \sigma_1$. We define the top-$k$ rows subspace of $X$ as the subspace spawned by the first $k$ columns of $V$.

**Fact B.1** (Min-Max principle for singular values)**.** *For every matrix $X \in \mathbb{R}^{n \times d}$ and $i \in [\min\{n, d\}]$ it holds that*

$$\sigma_i(X) = \max_{\dim(\mathcal{E})=i} \min_{v \in \mathcal{S}_d \cap \mathcal{E}} \|Xv\|_2 = \min_{\dim(\mathcal{E})=i} \max_{v \in \mathcal{S}_d \cap \mathcal{E}} \|Xv\|_2.$$

## B.4   Concentration Bounds

**Fact B.2** (Montgomery-Smith [1990])**.** *Let* $\mathbf{x} = (\mathbf{x}_1, \ldots, \mathbf{x}_n)$ *where the* $\mathbf{x}_i$*'s are i.i.d. random variables with* $\Pr[\mathbf{x}_i = 1] = \Pr[\mathbf{x}_i = -1] = 1/2$*, and let* $v \in \mathbb{R}^n$*. Then*

$$\forall t \geq 0 : \quad \Pr[|\langle \mathbf{x}, v \rangle| > t \cdot \|v\|_2] \leq 2\exp(-t^2/2).$$

**Fact B.3** (Bernstein's Inequality for sampling without replacement (Bardenet and Maillard [2015], Proposition 1.4))**.** *Let* $\mathbf{x}_1, \ldots, \mathbf{x}_m$ *be a random sample drawn without replacement from* $\{w_1, \ldots, w_n\}$ *for* $n \geq m$*. Let* $a = \min_{i \in [n]} w_i$*,* $b = \max_{i \in [n]} w_i$*,* $\mu = \frac{1}{n}\sum_{i=1}^n w_i$ *and* $\sigma^2 = \frac{1}{n}\sum_{i=1}^n (w_i - \mu)^2$*. Then for every* $t \geq 0$*,*

$$\Pr\left[\left|\sum_{i=1}^m \mathbf{x}_i - m \cdot \mu\right| \geq t\right] \leq 2 \cdot \exp\left(-\frac{t^2}{m\sigma^2 + (b-a)t/3}\right).$$

The following proposition is used for proving Lemma 2.1.

**Proposition B.4.** *Let* $X = (x_1, \ldots, x_n) \in (\mathcal{S}_d)^n$ *with singular values* $\sigma_1 \geq \ldots \geq \sigma_{\min\{n,d\}} \geq 0$*. Let* $\mathbf{X}' = (\mathbf{x}_1', \ldots, \mathbf{x}_m')$ *be the random matrix that is generated by taking a uniformly random*

$m$-size subset of the rows of $X$. Let $v \in \mathcal{S}_d$ and $a^2 = \|Xv\|_2^2 = \sum_{i=1}^n \langle x_i, v \rangle^2$. Assuming that $m \geq \frac{2n \ln(1/\beta)}{9a^2}$, it holds that

$$\Pr\left[\left| \|\mathbf{X}'v\|_2^2 - \frac{m}{n}a^2 \right| \geq \sqrt{2\frac{m}{n}a^2 \ln(1/\beta)} \right] \leq 2\beta.$$

*Proof.* The random variable $\|\mathbf{X}'v\|_2^2$ is distributed as the sum of a uniformly random $m$-size subset of $(\langle x_1, v \rangle^2, \dots, \langle x_n, v \rangle^2)$ (each element is bounded in $[0, 1]$). In the notations of Fact B.3, it holds that $\mu = \frac{1}{n} \sum_{i=1}^n \langle x_i, v \rangle^2 = a^2/n$, and

$$\begin{aligned}
\sigma^2 &= \frac{1}{n} \sum_{i=1}^n (\langle x_i, v \rangle^2 - a^2/n)^2 \\
&= \frac{1}{n} \sum_{i=1}^n (\langle x_i, v \rangle^4 - 2\langle x_i, v \rangle^2 a^2/n + a^4/n^2) \\
&= \frac{1}{n} \sum_{i=1}^n \langle x_i, v \rangle^4 - a^4/n^2 \\
&\leq \frac{1}{n} \sum_{i=1}^n \langle x_i, v \rangle^2 = a^2/n,
\end{aligned}$$

where the last inequality holds since $\langle x_i, v \rangle^2 \leq 1$.

Let $t = \sqrt{2\frac{m}{n}a^2 \ln(1/\beta)}$. By Fact B.3,

$$\begin{aligned}
\Pr\left[\left| \|\mathbf{X}'v\|_2^2 - \frac{m}{n}a^2 \right| \geq t \right] &\leq 2 \cdot \exp\left(-\frac{t^2}{\frac{m}{n}a^2 + t/3}\right) \\
&= 2 \cdot \exp\left(-\frac{2\frac{m}{n}a^2 \ln(1/\beta)}{\frac{m}{n}a^2 + \sqrt{2\frac{m}{n}a^2 \ln(1/\beta)}/3}\right) \\
&\leq 2 \cdot \exp\left(-\frac{2\frac{m}{n}a^2 \ln(1/\beta)}{2\frac{m}{n}a^2}\right) \\
&\leq 2\beta,
\end{aligned}$$

where the penultimate inequality holds by the assumption on $m$. $\square$

### B.4.1 Hypergeometric Distributions

**Definition B.5.** *For $n \in \mathbb{N}$, $m \in [n]$ and $w \in \{-n, \dots, n\}$, define the* Hypergeometric *probability distribution $\mathcal{HG}_{n,m,w}$ as the output of the following process: Take a vector $v \in \{-1, 1\}^n$ with $\sum_{i=1}^n v_i = w$, choose a uniformly random subset $\mathcal{I} \subseteq [n]$ of size $m$, and output $\sum_{i \in \mathcal{I}} v_i$.*

**Fact B.6** (Scala [2009], Equations 10 and 14). *If $\mathbf{x} \sim \mathcal{HG}_{n,m,w}$ then*

$$\forall t \geq 0: \quad \Pr[|\mathbf{x} - \mu| \geq t] \leq e^{-\frac{t^2}{2\ell}},$$

*where $\mu = \mathrm{E}[\mathbf{x}] = \frac{m \cdot w}{n}$.*

### B.4.2 Sub-Exponential Distributions

**Definition B.7** (Sub-Exponential Random Variable and Norm). *We say that a random variable $\mathbf{x} \in \mathbb{R}$ is* sub-exponential *if there exists $t > 0$ such that $\mathrm{E}\left[e^{|\mathbf{x}|/t}\right] \leq 2$. The* sub-exponential *norm of $\mathbf{x}$, denoted $\|\mathbf{x}\|_{\psi_1}$, is*

$$\|\mathbf{x}\|_{\psi_1} = \inf\{t > 0: \mathrm{E}\left[e^{|\mathbf{x}|/t}\right] \leq 2\}.$$

**Fact B.8** (Bernstein's inequality (Theorem 2.8.1 in Vershynin [2018])). *Let $\mathbf{x}_1, \dots, \mathbf{x}_n$ be independent, mean zero, sub-exponential random variables. Then*

$$\forall t \geq 0: \quad \Pr\left[\left| \sum_{i=1}^n \mathbf{x}_i \right| \geq t \right] \leq 2 \exp\left(-\Omega\left(\min\left(\frac{t^2}{\sum_{i=1}^n \|\mathbf{x}_i\|_{\psi_1}^2}, \frac{t}{\max_i \|\mathbf{x}_i\|_{\psi_1}}\right)\right)\right).$$

## B.5 Nets

**Definition B.9** ($\gamma$-Net)**.** *Let $\mathcal{T}$ be a subspace of $\mathbb{R}^d$. Consider a subset $\mathcal{K} \subset T$ and let $\gamma > 0$. A subset $\mathcal{N} \subseteq \mathcal{K}$ is called an $\gamma$-net of $\mathcal{K}$ if every point in $\mathcal{K}$ is within distance $\gamma$ of some point of $\mathcal{N}$, i.e.,*

$$\forall x \in \mathcal{K} \, \exists y \in \mathcal{N} \; : \; \|x - y\|_2 \leq \gamma$$

**Fact B.10** (Extension of Corollary 4.2.13 in Vershynin [2018])**.** *If $E$ is a subspace with $\dim(E) = k$, then there exists an $\gamma$-net of size $(3/\gamma)^k$ to the unit sphere in $E$ (i.e., to $E \cap \mathcal{S}_d$).*

## B.6 Projections

Recall that $\mathcal{W}_{d,k}$ denotes the set of all $d \times d$ rank-$k$ projection matrices. For a matrix $A \in \mathbb{R}^d$, we denote that $\Pi_A$ the projection matrix onto the subspace spawned by the columns of $A$ (in case $A$ is a unitary matrix, $\Pi_A = AA^T$).

**Fact B.11** (Theorem 1 and Lemma 1 in Cai and Zhang [2018])**.** *Let $X, Y, Z \in \mathbb{R}^{n \times d}$ such that $X = Y + Z$. Let $[U \, U_\perp]\Sigma[V \, V_\perp]^T$ be the SVD of $Y$, and let $[\widehat{U} \, \widehat{U}_\perp]\widehat{\Sigma}[\widehat{V} \, \widehat{V}_\perp]^T$ be the SVD of $X$, where $U, V, \widehat{U}, \widehat{V}$ denote the first $k$ columns of $[U \, U_\perp], [V \, V_\perp], [\widehat{U} \, \widehat{U}_\perp], [\widehat{V} \, \widehat{V}_\perp]$ (respectively). Let $Z_{12} = \Pi_U Z \Pi_{V_\perp}$ and $Z_{21} = \Pi_{U_\perp} Z \Pi_V$. In addition, let $z_{ij} = \|Z_{ij}\|$, let $\alpha = \sigma_{\min}(U^T X V)$ (i.e., the smallest singular value larger than 0), and $\beta = \left\|U_\perp^T X V_\perp\right\|$. If $\alpha^2 > \beta^2 + \min\{z_{12}^2, z_{21}^2\}$, then*

$$\left\|\Pi_V - \Pi_{\widehat{V}}\right\| \leq 2 \cdot \frac{\alpha z_{12} + \beta z_{21}}{\alpha^2 - \beta^2 - \min\{z_{12}^2, z_{21}^2\}}$$

*and*

$$\left\|\Pi_V - \Pi_{\widehat{V}}\right\|_F \leq \sqrt{2} \cdot \frac{\alpha\|Z_{12}\|_F + \beta\|Z_{21}\|_F}{\alpha^2 - \beta^2 - \min\{z_{12}^2, z_{21}^2\}}.$$

**Proposition B.12.** *Let $X, Y, Z, V, \widehat{V}$ as in Fact B.11 such that $Y$ has rank-$k$ and $Span(Y^T)$ is orthogonal to $Span(Z^T)$. If $\sigma_k(Y)^2 \geq 2\|Z\|^2$, then*

1. $\left\|\Pi_V - \Pi_{\widehat{V}}\right\| \leq 4 \cdot \frac{\|Z\|}{\sigma_k(Y)}$, *and*

2. $\left\|\Pi_V - \Pi_{\widehat{V}}\right\|_F \leq 2\sqrt{2} \cdot \frac{\|Z\|_F}{\sigma_k(Y)}$.

*Proof.* Note that $Span(V) = Span(Y^T)$. Therefore $ZV = 0$, which implies that $Z_{21} = \Pi_{U_\perp} Z \Pi_V = 0$. Compute

$$\alpha = \sigma_{\min}(U^T X V)$$
$$= \sigma_{\min}(U^T Y V + U^T \underbrace{ZV}_{0})$$
$$= \sigma_k(Y).$$

$$\beta = \left\|U_\perp^T X V_\perp\right\|$$
$$= \left\|U_\perp^T \underbrace{Y V_\perp}_{0} + U_\perp^T Z V_\perp\right\|$$
$$\leq \|U_\perp\| \cdot \|Z\| \cdot \|V_\perp\|$$
$$\leq \|Z\|.$$

$$\|Z_{12}\| = \|\Pi_U \cdot Z \cdot \Pi_{V_\perp}\| \leq \|Z\|,$$

$$\|Z_{12}\|_F \leq \|\Pi_U \cdot Z \cdot \Pi_{V_\perp}\|_F \leq \|Z\|_F,$$

where the last inequalities in the above two equations hold since for any matrix $A$ and projection matrices $\Pi_1, \Pi_2$ it holds that $\|\Pi_1 A \Pi_2\| \leq \|A\|$ and $\|\Pi_1 A \Pi_2\|_F \leq \|A\|_F$. The proof now immediately follows by applying Fact B.11. $\square$

**Proposition B.13.** *Let $P \in \mathbb{R}^{n \times d}$ be a rank-$k$ matrix and let $P' \in \mathbb{R}^{n \times d}$ such that $\|P - P'\|_F \leq \alpha$. Let $\Pi$ be the projection matrix onto $Span(P^T)$, and let $\Pi'$ be the projection onto the top-$k$ rows subspace of $P'$. If $\sigma_k(P) \geq 2\alpha$, then*

$$\|\Pi - \Pi'\|_F \leq 2\sqrt{2} \cdot \frac{\alpha}{\sigma_k(P) - \alpha}.$$

*Proof.* Let $E = P' - P$, and divide $E$ into $E = E_P + E_{\bar{P}}$ where the rows of $E_P$ belong to the rows subspace of $P$ and the rows of $E_{\bar{P}}$ are orthogonal to it. Let $Y = P + E_P$, so we can write $P' = Y + E_{\bar{P}}$. Note that $\|E_{\bar{P}}\|_F \leq \alpha$, and $\sigma_k(Y) \geq \sigma_k(P) - \|E_P\| \geq \sigma_k(P) - \alpha$. The proof now follows by applying Proposition B.12(2) on $P', Y, E_{\bar{P}}$. $\qquad\square$

**Fact B.14** (Implied by Corollary 4.6 in Singhal and Steinke [2021]). *Let $\Pi, \Pi_1, \ldots, \Pi_t \in \mathcal{W}_{d,k}$ s.t. for all $j \in [t]$, $\|\Pi - \Pi_j\| \leq \alpha$. Let $\mathbf{p}_1, \ldots, \mathbf{p}_q$ be i.i.d. random vectors in $\mathbb{R}^d$ from $\mathcal{N}(\vec{0}, \mathbb{I}_{d \times d})$. Then*

$$\Pr\left[\forall i \in [q], j \in [t], \, \|(\Pi - \Pi_j)p_i\|_2 \leq O\left(\alpha\left(\sqrt{k} + \sqrt{\ln(qt)}\right)\right)\right] \geq 0.95.$$

**Fact B.15** (Implied by the proof of Lemma 4.9 in Singhal and Steinke [2021]). *Let $\Pi \in \mathcal{W}_{d,k}$ and let $\mathbf{p}_1, \ldots, \mathbf{p}_q$ be i.i.d. random vectors in $\mathbb{R}^d$ from $\mathcal{N}(\vec{0}, I_{d \times d})$. Let $\mathbf{P}$ be the $d \times q$ matrix whose columns are $\Pi\mathbf{p}_1, \ldots, \Pi\mathbf{p}_q$. If $q \geq c \cdot k$ for some large enough constant $c$, then w.p. $0.95$ it hold that $\sigma_k(\mathbf{P}) \geq \Omega(\sqrt{k})$ (and in particular, $Span(\mathbf{P}) = Span(\Pi)$).*

**Proposition B.16.** *For any $\Pi, \widetilde{\Pi} \in \mathcal{W}_{d,k}$ and $X \in (\mathcal{S}_d)^n$, it holds that*

$$\left\|\Pi \cdot X^T\right\|_F^2 - \left\|\widetilde{\Pi} \cdot X^T\right\|_F^2 \leq 2n \cdot \left\|\Pi - \widetilde{\Pi}\right\|_F.$$

*Proof.* Compute

$$
\begin{aligned}
\left\|\Pi X^T\right\|_F^2 - \left\|\widetilde{\Pi} X^T\right\|_F^2 &= (\left\|\Pi X^T\right\|_F - \left\|\widetilde{\Pi} X^T\right\|_F) \cdot (\left\|\Pi X^T\right\|_F + \left\|\widetilde{\Pi} X^T\right\|_F) \\
&\leq (\left\|\Pi X^T\right\|_F - \left\|\widetilde{\Pi} X^T\right\|_F) \cdot 2\sqrt{n} \\
&\leq \left\|(\Pi - \widetilde{\Pi}) X^T\right\|_F \cdot 2\sqrt{n} \\
&\leq \left\|\Pi - \widetilde{\Pi}\right\|_F \cdot \left\|X^T\right\|_F \cdot 2\sqrt{n} \\
&= 2n \cdot \left\|\Pi - \widetilde{\Pi}\right\|_F.
\end{aligned}
$$

$\qquad\square$

**Proposition B.17.** *Let $v_1, \ldots, v_k \in \mathcal{S}_d$ with $\max_{i,j}|\langle v_i, v_j \rangle| \leq \alpha \leq \frac{1}{20}$. Let $u_1, \ldots, u_k$ be the result of the Gram-Schmidt process applied on $v_1, \ldots, v_k$. Then for every $i \in [k]$, there exists $\lambda_{i-1} \in \mathbb{R}$ with $|\lambda_{i-1}| \leq \alpha(1 + 4\alpha)$ and $w_i \in \mathbb{R}^d$ with $\|w_i\|_2 \leq 2\alpha^2$ such that*

$$u_i = v_i + \lambda_{i-1} v_{i-1} + w_i.$$

*Proof.* We prove it by induction on $i$. The case $i = 1$ holds since $u_1 = v_1$. Assume it holds for $i$, and we prove it for $i + 1$. Define $\lambda_i = -\langle u_i, v_{i+1} \rangle$ and $w_{i+1} = \lambda_i(\lambda_{i-1} v_{i-1} + w_i)$. Note that

$$
\begin{aligned}
|\lambda_i| &= |\langle v_i + \lambda_{i-1} v_{i-1} + w, \, v_{i+1} \rangle| \\
&= |\langle v_i, v_{i+1} \rangle + \lambda_{i-1}\langle v_{i-1}, v_{i+1} \rangle + \langle w, v_{i+1} \rangle| \\
&\leq \alpha + |\lambda_{i-1}|\alpha + \|w\|_2 \\
&\leq \alpha + \alpha^2(1 + 4\alpha) + 2\alpha^2 \\
&\leq \alpha(1 + 4\alpha),
\end{aligned}
$$

and that $\|w\|_2 \leq \alpha^2(1 + 4\alpha)^2 + 2\alpha^3 \leq 2\alpha^2$ (recall that $\alpha \leq \frac{1}{20}$). The proof now follows since

$$
\begin{aligned}
u_{i+1} &= v_{i+1} - \langle u_i, v_{i+1} \rangle u_i \\
&= v_{i+1} + \lambda_i(v_i + \lambda_{i-1} v_{i-1} + w_i) \\
&= v_{i+1} + \lambda_i v_i + w_{i+1}.
\end{aligned}
$$

$\qquad\square$

### B.7 Algorithms

Let M be a randomized algorithm that uses $m$ random coins. For $r \in \{0,1\}^m$ we denote by $M_r$ the (deterministic) algorithm M after fixing its random coins to $r$. Given an oracle-aided algorithm A and algorithm B, we denote by $A^B$ the algorithm A with oracle access to B.

### B.8 Differential Privacy

**Definition B.18** (Differential privacy [Dwork et al., 2006b,a]). *A randomized mechanism* $M\colon \mathcal{X}^n \to \mathcal{Y}$ *is* $(\varepsilon, \delta)$*-differentially private (in short,* $(\varepsilon, \delta)$*-DP) if for every neighboring databases* $X = (x_1, \ldots, x_n)$, $X' = (x'_1, \ldots, x'_n) \in \mathcal{X}^n$ *(i.e., differ by exactly one entry), and every set of outputs* $\mathcal{T} \subseteq \mathcal{Y}$*, it holds that*

$$\Pr[M(X) \in \mathcal{T}] \le e^\varepsilon \cdot \Pr[M(X') \in \mathcal{T}] + \delta$$

#### B.8.1 Basic Facts

**Fact B.19** (Post-Processing). *If* $M\colon \mathcal{X}^n \to \mathcal{Y}$ *is* $(\varepsilon, \delta)$*-DP then for every randomized* $F\colon \mathcal{Y} \to \mathcal{Z}$*, the mechanism* $F \circ M\colon \mathcal{X}^n \to \mathcal{Z}$ *is* $(\varepsilon, \delta)$*-DP.*

Post-processing holds when applying the function on the output of the DP mechanism. In this work we sometimes need to apply the mechanism on the output of a function. While this process does not preserve DP in general, it does so assuming the function is *neighboring-preserving*.

**Definition B.20** (Neighboring-Preserving Algorithm). *We say that a randomized algorithm* $G\colon \mathcal{X}^n \to \mathcal{Y}^m$ *is* neighboring-preserving *if for every neighboring* $X, X' \in \mathcal{X}^n$*, the outputs* $G(X), G(X') \in \mathcal{Y}^m$ *are neighboring w.p.* 1.

**Fact B.21.** *If* $G\colon \mathcal{X}^n \to \mathcal{Y}^m$ *is neighboring-preserving and* $M\colon \mathcal{Y}^m \to \mathcal{Z}$ *is* $(\varepsilon, \delta)$*-DP, then* $M \circ G\colon \mathcal{X}^n \to \mathcal{Z}$ *is* $(\varepsilon, \delta)$*-DP.*

#### B.8.2 Zero-Concentrated Differential Privacy (zCDP)

Our empirical evaluation (Section 4) is performed in the zCDP model of Bun and Steinke [2016], defined below.

**Definition B.22** (Rényi Divergence (Rényi [1961])). *Let* $\mathbf{y}$ *and* $\mathbf{y}'$ *be random variables over* $\mathcal{Y}$*. For* $\alpha \in (1, \infty)$*, the* Rényi divergence *of order* $\alpha$ *between* $\mathbf{y}$ *and* $\mathbf{y}'$ *is defined by*

$$D_\alpha(\mathbf{y}\|\mathbf{y}') = \frac{1}{\alpha - 1} \cdot \ln\left( \mathrm{E}_{y \leftarrow \mathbf{y}}\left[ \left( \frac{P(y)}{P'(y)} \right)^{\alpha - 1} \right] \right),$$

*where* $P(\cdot)$ *and* $P'(\cdot)$ *are the probability mass/density functions of* $\mathbf{y}$ *and* $\mathbf{y}'$*, respectively.*

**Definition B.23** (zCDP Indistinguishability). *We say that two random variable* $\mathbf{y}, \mathbf{y}'$ *over a domain* $\mathcal{Y}$ *are* $\rho$*-indistinguishable (denote by* $\mathbf{y} \approx_\rho \mathbf{y}'$*), iff for every* $\alpha \in (1, \infty)$ *it holds that*

$$D_\alpha(\mathbf{y}\|\mathbf{y}'), D_\alpha(\mathbf{y}'\|\mathbf{y}) \le \rho\alpha.$$

*We say that* $\mathbf{y}, \mathbf{y}'$ *are* $(\rho, \delta)$*-indistinguishable (denote by* $\mathbf{y} \approx_{\rho,\delta} \mathbf{y}'$*), iff there exist events* $E, E' \subseteq \mathcal{X}$ *with* $\Pr[\mathbf{y} \in E], \Pr[\mathbf{y}' \in E'] \ge 1 - \delta$ *such that* $\mathbf{y}|_E \approx_\rho \mathbf{y}|_{E'}$*.*

**Definition B.24** ($(\rho, \delta)$-zCDP Bun and Steinke [2016]). *A mechanism* M *is* $\delta$*-approximate* $\rho$*-zCDP (in short,* $(\rho, \delta)$*-zCDP), if for any neighboring databases* $X, X'$ *it holds that* $M(X) \approx_{\rho,\delta} M(X')$*.*

#### The Gaussian Mechanism

**Fact B.25** (The Gaussian Mechanism Dwork et al. [2006a], Bun and Steinke [2016]). *Let* $\boldsymbol{x}, \boldsymbol{x}' \in \mathbb{R}^d$ *be vectors with* $\|\boldsymbol{x} - \boldsymbol{x}'\|_2 \le \lambda$*. For* $\rho > 0$*,* $\sigma = \frac{\lambda}{\sqrt{2\rho}}$ *and* $Z \sim \mathcal{N}(\mathbf{0}, \sigma^2 \cdot \mathbb{I}_{d \times d})$ *it holds that* $\boldsymbol{x} + Z \approx_\rho \boldsymbol{x}' + Z$*.*

#### B.8.3 FriendlyCore Averaging

We use the following DP averaging algorithm that given the diameter $\xi$ of a ball that contain most of the points, it can estimate their average.

**Fact B.26** (Tsfadia et al. [2022]). *Let $\lambda \geq 1$ and $\delta \leq \varepsilon, \beta \leq 1$. There exists an $(\varepsilon, \delta)$-DP algorithm* FC_Average *that gets as input a dataset $S = (x_1, \ldots, x_n) \in (\mathbb{R}^d)^n$ and a parameter $\xi > 0$ and satisfies the following utility guarantee: If $n \geq O\left( \frac{\log(1/\delta)}{\varepsilon} + \frac{\sqrt{d\log(1/\delta)\log(1/\beta)}}{\lambda\varepsilon} \right)$ and $\exists S' \subseteq S$ with $|S'| \geq 0.8n$ s.t. $\forall x_i, x_j \in S' : \|x_i - x_j\|_2 \leq \xi$, then*

$$\Pr_{y \sim \textsf{FC\_Average}(S, \xi)}[\|y - \mu\| > \lambda\xi] \leq \beta,$$

*where $\mu = \frac{1}{|S'|}\sum_{x \in S'} x$. Furthermore, the running time of* FC_Average$(S, \cdot)$ *is $\tilde{O}(dn\log(n/\delta))$ (See Appendix B in Tsfadia et al. [2022]).*

Fact B.26 is not explicitly stated in Tsfadia et al. [2022] since they only analyzed the utility guarantee of their averaging in the zCDP model. Yet, it can be achieved using similar steps. First, we need to consider a "friendly" DP variant of their FriendlyAvg algorithm (Algorithm 3.3 in Tsfadia et al. [2022]), and as Tsfadia et al. [2022] noted, we can do it such that the probability of failure is low whenever $n \geq O\left( \frac{\log(1/\delta)}{\varepsilon} \right)$, and the additive error (given success) decreases in a rate of $\frac{\xi\sqrt{d\log(1/\delta)\log(1/\beta)}}{n\varepsilon}$, where $\xi$ is the diameter of the points. Fact B.26 immediately obtained by combining FriendlyAvg with their paradigm for DP (Theorem 4.11 in Tsfadia et al. [2022] applied with $\alpha = 0.2$).

### B.8.4 Subspace Recovery Algorithm of Dwork et al. [2014]

We next describe the subspace recovery algorithm of Dwork et al. [2014] that strongly takes advantage of a large additive gap $\sigma_k^2 - \sigma_{k+1}^2$ for decreasing the noise that is required for privacy. This algorithm is only used in our empirical evaluation (Section 4).

---

**Algorithm B.27** (Algorithm 2 in Dwork et al. [2014]).
*Input: A dataset $X = (x_1, \ldots, x_n) \in (\mathcal{S}_d)^n$.*
*DP parameters: $\varepsilon, \delta$.*
*Rank parameter: $k$.*
*Operation:*

1. *Compute a projection matrix $\Pi$ to the top-$k$ rows subspace of $X$, and compute the singular values $\sigma_k, \sigma_{k+1}$.*
2. *Compute $g = \sigma_k^2 - \sigma_{k+1}^2 + Lap(2/\varepsilon)$.*
3. *Compute $W = \Pi + E$, where $E$ is a $d \times d$ symmetric matrix where the upper triangle is i.i.d. samples from $\mathcal{N}(\mathbf{0}, \left( \frac{\Delta_{\varepsilon,\delta}}{g - 2\log(1/\delta)/\varepsilon - 2} \right)^2)$, where $\Delta_{\varepsilon,\delta} = \frac{1 + \sqrt{2\log(1/\delta)}}{\varepsilon}$.*
4. *Output a projection matrix $\tilde{\Pi}$ to the top $k$ eigenvectors of $W$.*

---

Note that when the additive gap $\sigma_k^2 - \sigma_{k+1}^2$ is large, the algorithm will add smaller noise per coordinate in Step 3.

**Fact B.28** (Theorem 11 in Dwork et al. [2014]). *Algorithm B.27 is $(2\varepsilon, 2\delta)$-DP.*

The privacy analysis is done by the following steps. First, by the Laplace mechanism, Step 2 is $\varepsilon$-DP. Second, by tail bound on the Laplace distribution, the probability that $g - 2\log(1/\delta)/\varepsilon \leq \sigma_k^2 - \sigma_{k+1}^2$ is at least $1 - \delta$. Furthermore, they show that if $\sigma_k^2 - \sigma_{k+1}^2 \geq \alpha$ then the Forbenius-norm sensitivity of the matrix $\Pi$ is at most $\frac{2}{\alpha - 2}$. So conditioned on the above $1 - \delta$ probability event, the Forbenius-norm sensitivity of $\Pi$ is at most $\frac{2}{g - 2\log(1/\delta)/\varepsilon - 2}$, and therefore the Gaussian mechanism step (Item 3) guarantees $(\varepsilon, \delta)$-DP, and by composition the entire process is $(2\varepsilon, 2\delta)$-DP.

In order to consider a zCDP version of Algorithm B.27, we replace the Laplace noise $Lap(2/\varepsilon)$ with a Gaussian noise $\mathcal{N}(0, \sigma^2)$ for $\sigma = \sqrt{2/\rho}$. Given this change, now it holds that $g - \sigma\sqrt{2\ln(1/\delta)} \leq \sigma_k^2 - \sigma_{k+1}^2$ w.p. at least $1 - \delta$ (by tail bound on Gaussian distribution). Finally, in Step 3 we replace $\Delta_{\varepsilon,\delta}$ (the required standard deviation for $(\varepsilon, \delta)$-DP) to $1/\sqrt{2\rho}$ (what is required for $\rho$-zCDP). This results with the following $(2\rho, \delta)$-zCDP algorithm:

**Algorithm B.29** (zCDP version of Algorithm 2 in Dwork et al. [2014]).

*Input: A dataset $X = (x_1, \ldots, x_n) \in (\mathcal{S}_d)^n$.*

*zCDP parameters: $\rho, \delta$.*

*Rank parameter: $k$.*

*Operation:*

1. *Compute a projection matrix $\Pi$ to the top-$k$ rows subspace of $X$, and compute the singular values $\sigma_k, \sigma_{k+1}$.*
2. *Compute $g = \sigma_k^2 - \sigma_{k+1}^2 + \mathcal{N}(0, 2/\rho)$.*
3. *Compute $W = \Pi + E$, where $E$ is a $d \times d$ symmetric matrix where the upper triangle is i.i.d. samples from $\mathcal{N}(\mathbf{0}, \left( \frac{\sqrt{1/(2\rho)}}{g - 2\sqrt{\ln(1/\delta)/\rho} - 2} \right)^2)$.*
4. *Output a projection matrix $\tilde{\Pi}$ to the top $k$ eigenvectors of $W$.*

### B.8.5   Lower Bounding Tools from Peter et al. [2024]

Peter et al. [2024] showed that for $d = \tilde{\Theta}(n^2)$, the distribution $\mathcal{D}(n, d)$ below induces a fingerprinting codebook for $n$ users, each codeword is of length $d$.

**Definition B.30** (FPC hard distribution $\mathcal{D}(n, d)$ Peter et al. [2024]). *Let $\rho$ be the distribution that outputs $p = (e^t - 1)/(e^t + 1) \in [-1, 1]$ for $t \leftarrow [-\ln(5n), \ln(5n)]$. For $n, d \in \mathbb{N}$, let $\mathcal{D}(n, d)$ be the distribution that chooses independently $p^1, \ldots, p^d \sim \rho$, and outputs a codebook $(x_1, \ldots, x_n) \in (\{-1, 1\}^d)^n$ where for each $i \in [n]$ and $j \in [d]$, $x_i^j$ is drawn independently over $\{-1, 1\}$ with expectation $p^j$.*

**Framework for Lower Bounds**   Consider a mechanism $\mathsf{M} \colon \mathcal{X}^n \to \mathcal{W}$ that satisfies some weak accuracy guarantee. Peter et al. [2024] showed that the task of proving a lower bound on $n$ is reduced to the following task: Transform an FPC codebook $X \in \{-1, 1\}^{n_0 \times d_0}$ into hard instances $Y \in \mathcal{X}^n$ for $\mathsf{M}$, and then extract from the output $w \in \mathcal{W}$ of $\mathsf{M}(Y)$ a vector $q \in \{-1, 1\}^{d_0}$ that is strongly-correlated with $X$ ($n_0$ and $d_0$ are some functions of $n$ and $d$ and the weak accuracy guarantee of $\mathsf{M}$), where

**Definition B.31** (Strongly Correlated). *We say that a random variable $\mathbf{q} = (\mathbf{q}^1, \ldots, \mathbf{q}^d) \in \{-1, 1\}^d$ is strongly-correlated with a matrix $X \in \{-1, 1\}^{n \times d}$, if*

$$\forall b \in \{-1, 1\}, \ \forall j \in \mathcal{J}_X^b : \quad \Pr\big[\mathbf{q}^j = b\big] \geq 0.9$$

*(recall that $\mathcal{J}_X^b = \{j \in [d] \colon x_i^j = b \text{ for all } i \in [n]\}$).*

Denote by $\mathsf{G} \colon \{-1, 1\}^{n_0 \times d_0} \times \mathcal{Z} \to \mathcal{X}^n$ the algorithm that generates the hard instances using a uniformly random secret $z \leftarrow \mathcal{Z}$ (i.e., $z$ could be a random permutation, a sequence of random permutations, etc). Denote by $\mathsf{F} \colon \mathcal{Z} \times \mathcal{W} \to \{-1, 1\}^{d_0}$ the algorithm that extracts a good $q$ using the secret $z$ and the output $w$. We denote by $\mathsf{A}^{\mathsf{M}, \mathsf{F}, \mathsf{G}}(X)$ the entire process:

**Definition B.32** (Algorithm $\mathsf{A}^{\mathsf{M}, \mathsf{F}, \mathsf{G}}$). *Let $\mathcal{Z}, \mathcal{W}$ be domains, and let $n_0, d_0, n, d \in \mathbb{N}$. Let $(\mathsf{M}, \mathsf{F}, \mathsf{G})$ be a triplet of randomized algorithms of types $\mathsf{G} \colon \{-1, 1\}^{n_0 \times d_0} \times \mathcal{Z} \to \mathcal{X}^n$, $\mathsf{M} \colon \mathcal{X}^n \to \mathcal{W}$, and $\mathsf{F} \colon \mathcal{Z} \times \mathcal{W} \to [-1, 1]^{d_0}$, each uses $m$ random coins. Define $\mathsf{A}^{\mathsf{M}, \mathsf{F}, \mathsf{G}} \colon \{-1, 1\}^{n_0 \times d_0} \to [-1, 1]^{d_0}$ as the randomized algorithm that on inputs $X \in \{-1, 1\}^{n_0 \times d_0}$, samples $z \leftarrow \mathcal{Z}$, $Y \sim \mathsf{G}(X, z)$ and $w \sim \mathsf{M}(Y)$, and outputs $q \sim \mathsf{F}(z, w)$.*

**Definition B.33** ($\beta$-Leaking). *Let $\mathsf{M}, \mathsf{F}, \mathsf{G}$ be randomized algorithms as in Definition B.32, each uses $m$ random coins, and let $\mathcal{D}(n_0, d_0)$ be the distribution from Definition B.30. We say that the triplet $(\mathsf{M}, \mathsf{F}, \mathsf{G})$ is $\beta$-leaking if*

$$\Pr_{r, r', r'' \leftarrow \{0, 1\}^m, \, X \leftarrow \mathcal{D}(n_0, d_0)}\big[\mathsf{A}^{\mathsf{M}_r, \mathsf{F}_{r'}, \mathsf{G}_{r''}}(X) \text{ is strongly-correlated with } X\big] \geq \beta,$$

*where recall that $\mathsf{M}_r$ denotes the algorithm $\mathsf{M}$ when fixing its random coins to $r$ ($\mathsf{F}_{r'}, \mathsf{G}_{r''}$ are similarly defined).*

**Lemma B.34** (Framework for Lower Bounds Peter et al. [2024]). *Let $\beta \in (0,1]$, $n_0, n, d_0, d \in \mathbb{N}$. Let* $\mathsf{M} \colon \mathcal{X}^n \to \mathcal{W}$ *be an algorithm such that there exists two algorithms* $\mathsf{G} \colon \{-1,1\}^{n_0 \times d_0} \times \mathcal{Z} \to \mathcal{X}^n$ *and* $\mathsf{F} \colon \mathcal{Z} \times \mathcal{W} \to [-1,1]^{d_0}$ *such that the triplet* $(\mathsf{M}, \mathsf{F}, \mathsf{G})$ *is $\beta$-leaking (Definition B.33). If* $\mathsf{M}$ *is* $\left(1, \frac{\beta}{4n_0}\right)$*-DP and* $\mathsf{G}(\cdot, z)$ *is neighboring-preserving (Definition B.20) for every* $z \in \mathcal{Z}$*, then* $n_0 \geq \Omega\left(\frac{\sqrt{d_0}}{\log^{1.5}(d_0/\beta)}\right)$.

**Padding-And-Permuting (PAP) FPC**   The main technical tool of Peter et al. [2024] for generating hard instance is to sample a random fingerprinting codebook from $\mathcal{D}(n, d_0)$, append many 1-marked and $(-1)$-marked columns, and randomly permute all the columns.

**Definition B.35** (PAP$_{n,d_0,\ell}$). *Let* $\ell, n, d_0 \in \mathbb{N}$*, and let* $d = d_0 + 2\ell$*. We define* PAP$_{n,d_0,\ell} \colon \{-1,1\}^{n \times d_0} \times \mathcal{P}_d \to \{-1,1\}^{n \times d}$ *as the function that given* $X \in \{-1,1\}^{n \times d_0}$ *and a permutation matrix* $P \in \mathcal{P}_d$ *as inputs, outputs* $X' = X'' \cdot P$ *(i.e., permutes the columns of* $X''$ *according to P), where* $X''$ *is the* $\{-1,1\}^{n \times d}$ *matrix after appending* $\ell$ *1-marked and* $\ell$ *$(-1)$-marked columns to* $X$ *(where recall that a b-marked is a column with all entries equal to b).*

Note that for every $n, d_0, \ell \in \mathbb{N}$ and $P \in \mathcal{P}_d$, the function PAP$_{n,d_0,\ell}(\cdot, P)$ is neighboring-preserving (Definition B.20).

**Definition B.36** (Strongly Agrees). *We say that a vector* $q = (q^1, \ldots, q^d)$ *strongly-agrees with a matrix* $X \in \{-1,1\}^{n \times d}$*, if*

$$\forall b \in \{-1,1\} \colon \quad \left|\{j \in \mathcal{J}_X^b \colon q^j = b\}\right| \geq 0.9\left|\mathcal{J}_X^b\right|.$$

The following lemma capture the main technical property of the PAP technique.

**Lemma B.37** (Peter et al. [2024]). *Let* $\ell, n, d_0 \in \mathbb{N}$ *such that* $d = d_0 + 2\ell$*. Let* $\mathsf{M} \colon \{-1,1\}^{n \times d} \to [-1,1]^d$ *be an mechanism that uses* $m$ *random coins,* $\mathbf{P} \leftarrow \mathcal{P}_d$ *(a random variable) and for* $X \in \{-1,1\}^{n \times d_0}$ *let* $\mathbf{Y}_X = \mathsf{PAP}(X, \mathbf{P})$*. Then for any distribution* $\mathcal{D}$ *over* $\{-1,1\}^{n \times d_0}$*:*

$$\Pr_{r \leftarrow \{0,1\}^m, \, X \sim \mathcal{D}}\left[(\mathsf{M}_r(\mathbf{Y}_X) \cdot \mathbf{P}^T)^{1,\ldots,d_0} \text{ is strongly-correlated with } X\right]$$
$$\geq \mathrm{E}_{X \sim \mathcal{D}}[\Pr[\mathsf{M}(\mathbf{Y}_X) \text{ strongly-agrees with } \mathbf{Y}_X]].$$

# C   Upper Bounds

In this section we prove our upper bounds for subspace estimation. In Appendix C.2 we prove Theorem 1.6, and in Appendix C.3 we prove Theorem 1.7. Both algorithm share a similar structure that is defined next in Appendix C.1.

## C.1   Base Algorithms

Similarly to Singhal and Steinke [2021], our algorithms will follow the sample and aggregate approach of Nissim et al. [2007]. That is, we partition the rows into $t$ subsets, compute (non-privately) the top-$k$ projection matrix of each subset, and then privately aggregate the projections. This is Algorithm C.1 that uses oracle access to an aggregation algorithm. Unlike Singhal and Steinke [2021] who assumed that the rows are i.i.d. Gaussian samples, here we take a *random* partition and show that with large enough probability over the randomness of the partition, the projection matrices are indeed close to each other. We consider two types of aggregations: The first type, called Algorithm C.2, simply treats the matrices as vectors of dimension $d^2$ and computes a DP-average of them using FriendlyCore averaging (Fact B.26). The second type, called Algorithm C.3, is more similar to the aggregation done by Singhal and Steinke [2021]. That is, sample reference points $p_1, \ldots, p_q$ and then aggregate the $kd$ dimensional points $\{(\Pi_j p_1, \ldots, \Pi_j p_q)\}_{j=1}^t$. The difference from Singhal and Steinke [2021] is that we use FriendlyCore averaging (Fact B.26) which simplifies the construction.

**Algorithm C.1** (Algorithm EstSubspace)**.**

*Input: A dataset $X = (x_1, \ldots, x_n) \in (\mathcal{S}_d)^n$.*

*Parameters: $k, t$.*

*Oracle: A DP algorithm Agg for aggregating projection matrices.*

*Operation:*

1. *Randomly split $X$ into $t$ subsets, each contains (at least) $m = \lfloor n/t \rfloor$ rows.
   Let $X_1, \ldots, X_t$ be the resulting subsets.*
2. *For each $j \in [t]$: Compute the projection matrix $\Pi_j$ of the top-$k$ rows subset of $X_j$.*
3. *Output $\widetilde{\Pi} \sim \mathsf{Agg}(\Pi_1, \ldots, \Pi_t)$.*

---

**Algorithm C.2** (Algorithm Naive_Agg)**.**

*Input: A dataset $\vec{\Pi} = (\Pi_1, \ldots, \Pi_t) \in (\mathcal{W}_{d,k})^t$.*

*Privacy parameters: $\varepsilon, \delta \le 1$.*

*Utility parameter: $\xi \in [0, 1]$.*

*Operation:*

1. *Compute $\widehat{\Pi} \sim \mathsf{FC\_Average}_{\varepsilon, \delta}(\vec{\Pi}, \xi)$ (i.e., each $\Pi_j$ is treated like a vector in $\mathbb{R}^{d^2}$).*
2. *Output $\widetilde{\Pi} = \arg\min_{\Pi \in \mathcal{W}_{d,k}} \left\| \Pi - \widehat{\Pi} \right\|_F$.*

---

**Algorithm C.3** (Algorithm SS_Agg)**.**

*Input: A dataset $(\Pi_1, \ldots, \Pi_t) \in (\mathcal{W}_{d,k})^t$.*

*Utility Parameters: $k, q, \xi$.*

*Privacy parameters: $\varepsilon, \delta \le 1$.*

*Operation:*

1. *Sample $p_1, \ldots, p_q \sim \mathcal{N}(\vec{0}, \mathbb{I}_{d \times d})$ (i.i.d. samples from a standard spherical Gaussian).*
2. *For $j \in [t]$, compute $y^j = (\Pi_j p_1, \ldots, \Pi_j p_q) \in \mathbb{R}^{qd}$, and let $Y = (y^1, \ldots, y^t)$.*
3. *Compute $z = (z_1, \ldots, z_{qd}) \sim \mathsf{FC\_Average}_{\varepsilon, \delta}(Y, \xi)$ ($z \in \mathbb{R}^{qd}$).*
4. *Let $\widetilde{P}$ be the $q \times d$ matrix whose $i^{\text{th}}$ row (for $i \in [q]$) is $(z_{d(i-1)+1}, \ldots, z_{di})$ (which estimates the projection of $p_i$ onto the top-$k$ rows subspace of $X$).*
5. *Output the projection matrix $\widetilde{\Pi}$ of the top-$k$ rows subspace of $\widetilde{P}$.*

### C.1.1  Running Time

We analyze the running time of $\mathsf{EstSubspace}_{k,t}^{\mathsf{SS\_Agg}_{k,q,\xi}}$. Denote by $T(n, d, k)$ the running time of computing a projection matrix to the top-$k$ row subspace of an $n \times d$ matrix. The running time of Step 2 in EstSubspace is $t \cdot T(n/t, d, k)$. The running time of SS_Agg is $O(dqtk)$ on Step 2, $O(dqt \log t)$ on Step 3 (Fact B.26), and $T(q, d, k)$ on Step 5. Overall it is $t \cdot T(n/t, d, k) + T(q, d, k) + O(dqt(\log t + k))$. For both our weak and strong estimators (described next) we use $n/t = \tilde{\Theta}(k)$ and $q = \tilde{O}(k)$, and therefore we obtain that the total running time is $\frac{n}{m} \cdot T(m, d, k) + \tilde{O}(dkn)$ for $m = n/t = \tilde{\Theta}(k)$.

### C.1.2  Key Property

In order to claim that the (non-private) projection matrices are close to each other, we use the following lemma which states that with high enough probability over a random subset, the top-$k$ projection matrix in the subset is close to the top-$k$ projection matrix of the entire matrix.

**Lemma C.4** (Restatement of Lemma 2.1)**.** *Let $X = (x_1, \ldots, x_n) \in (\mathcal{S}_d)^n$ with singular values $\sigma_1 \ge \ldots \ge \sigma_{\min\{n,d\}} \ge 0$ and $\sigma_k^2 \ge 0.01 n/k$. Let $\mathbf{X}' \in (\mathcal{S}_d)^m$ be a uniformly random $m$-size*

*subset of the rows of $X$ (without replacement). Let $\Pi$ and $\mathbf{\Pi}'$ be the projection matrices to the top-$k$ rows subspace of $X$ and $\mathbf{X}'$, respectively. Then the following holds for $\gamma_1 = \frac{\sigma_{k+1}}{\sigma_k}$ and $\gamma_2 = \frac{\sqrt{\sum_{i=k+1}^{\min\{n,d\}} \sigma_i^2}}{\sigma_k}$:*

1. *If $m \geq \max\{800k \ln\left(\frac{k}{4\beta}\right), 2\gamma_1 n\}$, then $\Pr\left[\|\Pi - \mathbf{\Pi}'\| \leq 4\sqrt{\frac{2n}{m}} \cdot \gamma_1\right] \geq 1 - \beta/2$.*

2. *If $m \geq 800k \ln\left(\frac{k}{4\beta}\right)$ and $\beta \geq 4\gamma_2^2$, then $\Pr\left[\|\Pi - \mathbf{\Pi}'\|_F \leq 4\sqrt{\frac{2}{\beta}} \cdot \gamma_2\right] \geq 1 - \beta$.*

*Proof.* We will prove each part of the lemma by applying Proposition B.12. In the following, let $X = \sum_{i=1}^{n} \sigma_i u_i v_i^T$ be the SVD of $X$, and note that we can write $\mathbf{X}' = (x_{\mathbf{i}_1}, \ldots, x_{\mathbf{i}_m})$ where $\{\mathbf{i}_1, \ldots, \mathbf{i}_m\}$ is a random subset of $[n]$ (without replacement). Let $Y = (y_1, \ldots, y_n) = \sum_{i=1}^{k} \sigma_i u_i v_i^T$ and let $Z = (z_1, \ldots, z_n) = \sum_{i=k+1}^{\min\{n,d\}} \sigma_i u_i v_i^T$, and note that $Span\{y_1, \ldots, y_n\} = Span\{v_1, \ldots, v_k\}$ is orthogonal to $Span\{z_1, \ldots, z_n\} = Span\{v_{k+1}, \ldots, v_{\min\{n,d\}}\}$. Furthermore, define the random matrices $\mathbf{Y}' = (y_{\mathbf{i}_1}, \ldots, y_{\mathbf{i}_m})$ and $\mathbf{Z}' = (z_{\mathbf{i}_1}, \ldots, z_{\mathbf{i}_m})$ and note that $\mathbf{X}' = \mathbf{Y}' + \mathbf{Z}'$.

First, by Proposition B.4 (applied on $v_1, \ldots, v_k$) and the assumptions on $m, \sigma_k$, it holds by the union bound that

$$\Pr\left[\sigma_k(\mathbf{Y}') \geq \sqrt{\frac{m}{2n}} \sigma_k\right] \geq 1 - \beta/2. \tag{1}$$

In the following we assume that the event in Equation (1) occurs. We first prove Item 1. Note that $\|\mathbf{Z}'\| \leq \|\mathbf{Z}\| \leq \sigma_{k+1} \leq \gamma_1 \sigma_k$ and that $\sigma_k(\mathbf{Y}')^2 \geq \frac{m}{2n}\sigma_k^2 \geq 2\gamma_1^2\sigma_k^2 \geq 2\|\mathbf{Z}'\|^2$ (the second inequality holds since $m \geq 4\gamma_1^2 n$). By applying Proposition B.12(1) on $\mathbf{X}', \mathbf{Y}', \mathbf{Z}'$ we conclude that

$$\|\Pi - \mathbf{\Pi}'\| \leq 4 \cdot \frac{\|\mathbf{Z}'\|}{\sigma_k(\mathbf{Y}')} \leq 4\sqrt{\frac{2n}{m}} \cdot \gamma_1.$$

We next focus on proving Item 2. Note that $\|Z\|_F^2 = \sum_{i=k+1}^{\min\{n,d\}} \sigma_i^2 = \gamma_2^2\sigma_k^2$, and that $\mathrm{E}\left[\|\mathbf{Z}'\|_F^2\right] = \frac{m}{n}\|Z\|_F^2 = \frac{m}{n} \cdot \gamma_2^2\sigma_k^2$. Therefore by Markov's inequality

$$\Pr\left[\|\mathbf{Z}'\|_F^2 \leq \frac{2m}{\beta n}\gamma_2^2\sigma_k^2\right] \geq 1 - \beta/2. \tag{2}$$

In the following we assume that the event in Equation (2) occurs. Note that $\sigma_k(\mathbf{Y}')^2 \geq \frac{m}{2n}\sigma_k^2 \geq \frac{2m}{\beta n}\gamma_2^2\sigma_k^2 \geq 2\|\mathbf{Z}'\|^2$ (the second inequality holds since $\beta \geq 4\gamma_2^2$). By applying Proposition B.12(2) on $\mathbf{X}', \mathbf{Y}', \mathbf{Z}'$ we conclude that

$$\|\Pi - \mathbf{\Pi}'\| \leq 2\sqrt{2} \cdot \frac{\|\mathbf{Z}'\|_F}{\sigma_k(\mathbf{Y}')} \leq 2\sqrt{2} \cdot \frac{\sqrt{\frac{2m}{\beta n}}\gamma_2\sigma_k}{\sqrt{\frac{m}{2n}}\sigma_k} \leq 4\sqrt{\frac{2}{\beta}} \cdot \gamma_2.$$

$\square$

## C.2 Weak Estimator

In this section, we prove Theorem 1.6, stated below.

**Theorem C.5** (Restatement of Theorem 1.6). *Let $n, k, d \in \mathbb{N}$, $\lambda > 0$, $\varepsilon, \delta \in (0, 1]$ where $k \leq \min\{n, d\}$. There exists an $(k, \lambda, \beta = 0.9, \gamma_{\max} = \Omega(\min\{\frac{1}{\lambda}, 1\}))$-weak subspace estimator $\mathsf{M}: (\mathcal{S}_d)^n \times [0, 1] \to \mathbb{R}^{d \times d}$ with*

$$n = O\left(k \log k \left(\frac{\log(1/\delta)}{\varepsilon} + \frac{\min\{k\sqrt{d}, d\}\sqrt{\log(1/\delta)}}{\lambda\varepsilon}\right)\right)$$

*such that $\mathsf{M}(\cdot, \gamma)$ is $(\varepsilon, \delta)$-DP for every $\gamma \in [0, 1]$.*

Theorem C.5 is an immediate corollary of the following Lemmas C.6 and C.7.

**Lemma C.6.** *Let* $t = c_1 \cdot \left( \frac{\log(1/\delta)}{\varepsilon} + \frac{d\sqrt{\log(1/\delta)\log(20)}}{\lambda\varepsilon} \right)$ *(where $c_1$ is the hidden constant in Fact B.26). Then for any $n \geq 800k\ln(25k) \cdot t$, the mechanism* $\mathsf{M} \colon (\mathcal{S}_d)^n \times [0,1] \to \mathcal{W}_{d,k}$ *defined by* $\mathsf{M}(X,\gamma) := \mathsf{EstSubspace}_{k,t}^{\mathsf{Naive\_Agg}_{\varepsilon,\delta,\xi=60\gamma}}(X)$ *is an* $(k,\lambda,\beta = 0.9, \gamma_{\max} = \frac{1}{20})$*-weak-subspace-estimator.*

*Proof.* Let $X \in (\mathcal{S}_d)^n$ with $\sqrt{\frac{\sum_{i=k+1}^{\min\{n,d\}} \sigma_i(X)^2}{\sigma_k(X)^2}} \leq \gamma \leq \gamma_{\max}$ and let $\Pi \in \mathcal{W}_{d,k}$ be the projection of the top-$k$ rows subspace of $X$. Consider a random execution of $\mathsf{M}(X)$. Let $\{\mathbf{\Pi}_j\}_{j=1}^t, \widehat{\mathbf{\Pi}}$ be (random variables of) the values of $\{\Pi_j\}_{j=1}^t, \widehat{\Pi}$ in the execution, and let $\widetilde{\mathbf{\Pi}}$ be the output. By Lemma C.4(2) (recall that $\gamma_{\max} \leq \frac{1}{20}$) and the union bound,

$$\forall j \in [t]: \quad \Pr[\|\Pi - \mathbf{\Pi}_j\| \leq 60\gamma] \geq 0.99, \tag{3}$$

Let $\mathbf{a}_j = \mathbb{1}\{\|\Pi - \mathbf{\Pi}_j\| \leq 60\gamma\}$ (indicator random variable) and let $\mathbf{a} = \sum_{j=1}^t \mathbf{a}_j$. By Equation (3) it holds that $\mathrm{E}[\mathbf{a}] \geq 0.99t$, and recall that $\mathbf{a} \leq t$. It follows that

$$\Pr[\mathbf{a} \geq 0.8t] \geq \frac{\mathrm{E}[\mathbf{a}] - 0.8t \cdot \Pr[\mathbf{a} < 0.8t]}{t} \geq 0.99 - 0.8(1 - \Pr[\mathbf{a} \geq 0.8t]) \implies \Pr[\mathbf{a} \geq 0.8t] \geq 0.95. \tag{4}$$

In the following we assume that the event $\mathbf{a} \geq 0.8t$ occurs. Let $\mathbf{J} = \{j \in [t] \colon \mathbf{a}_j = 1\}$. Note that our choice of $t$ satisfies

$$t \geq c' \cdot \left( \frac{\log(1/\delta)}{\varepsilon} + \frac{d\sqrt{\log(1/\delta)\log(100)}}{\left(\frac{\lambda}{500}\right) \cdot \varepsilon} \right).$$

where $c'$ denotes the constant from Fact B.26. Therefore we conclude by Fact B.26 (FriendlyCore averaging) that

$$\Pr\left[ \left\| \mathbf{\Pi} - \widehat{\mathbf{\Pi}} \right\|_F \leq \frac{\lambda\gamma}{4} \right] \geq 0.99.$$

The proof of the lemma now follows by Proposition B.16 since $\left\| \mathbf{\Pi} - \widetilde{\mathbf{\Pi}} \right\|_F \leq 2 \left\| \mathbf{\Pi} - \widehat{\mathbf{\Pi}} \right\|_F$. $\square$

**Lemma C.7.** *Let $c_1$, $c_2$, $c_3$ be the constants from Facts B.14 to B.15 (respectively), and let $c$ be a large enough constant. Let* $t = c \cdot \left( \frac{\log(1/\delta)}{\varepsilon} + \frac{\left(\sqrt{k}+\sqrt{\log\frac{dk\log(1/\delta)}{\lambda\varepsilon}}\right)\sqrt{kd\log(1/\delta)}}{\lambda\varepsilon} \right)$, $\eta = c_1 \cdot \left( \sqrt{k} + \sqrt{\ln(qt)} \right)$ *and* $q = c_2 \cdot k$. *Then for any* $n \geq 800k\ln(25k) \cdot t$, *the mechanism* $\mathsf{M} \colon (\mathcal{S}_d)^n \times [0,1] \to \mathcal{W}_{d,k}$ *defined by* $\mathsf{M}(X,\gamma) := \mathsf{EstSubspace}_{k,t}^{\mathsf{SS\_Agg}_{\varepsilon,\delta,k,q,\xi=120\eta\sqrt{k}\gamma}}(X)$ *is an* $(k,\lambda,\beta = 0.9, \gamma_{\max} = \Theta(\min\{\frac{1}{\lambda}, 1\}))$*-weak-subspace-estimator.*

*Proof.* Let $X \in (\mathcal{S}_d)^n$ with $\sqrt{\frac{\sum_{i=k+1}^{\min\{n,d\}} \sigma_i(X)^2}{\sigma_k(X)^2}} \leq \gamma \leq \gamma_{\max}$ and let $\Pi \in \mathcal{W}_{d,k}$ be the projection of the top-$k$ rows subspace of $X$. Consider a random execution of $\mathsf{M}(X)$. Let $\{\mathbf{\Pi}_j\}_{j=1}^t, \{\mathbf{p}_i\}_{i=1}^q, \{\mathbf{y}^j\}_{j=1}^t, \mathbf{z}, \widetilde{\mathbf{P}}$ be (random variables of) the values of $\{\Pi_j\}_{j=1}^t, \{p_i\}_{i=1}^q, \{y^j\}_{j=1}^t, z, \widetilde{P}$ in the execution, and let $\widetilde{\mathbf{\Pi}}$ be the output. As in the proof of Lemma C.6, let $\mathbf{a} = \sum_{j=1}^t \mathbf{a}_j$ where $\mathbf{a}_j = \mathbb{1}\{\|\Pi - \mathbf{\Pi}_j\| \leq 60\gamma\}$. Then Equation (4) imples that

$$\Pr[\mathbf{a} \geq 0.8t] \geq 0.95.$$

In the following we assume that the event $\mathbf{a} \geq 0.8t$ occurs. Let $\mathbf{J} = \{j \in [t] \colon \mathbf{a}_j = 1\}$. By Fact B.14 and the definition of $\eta$ we obtain that

$$\Pr\left[ \forall i \in [q], j \in \mathbf{J} : \; \|(\Pi - \mathbf{\Pi}_j)\mathbf{p}_i\|_2 \leq 60\eta\gamma \right] \geq 0.95,$$

In the following we assume that the above event occurs. This yields that

$$\forall i, j \in \mathbf{J} : \ \|\mathbf{y}_i - \mathbf{y}_j\|_2 \leq 120\eta\sqrt{k}\gamma = \xi.$$

Furthermore, by the definition of $t$ and $\eta$ (note that $\eta$ depends on $\log(t)$), it holds that

$$t \geq c' \cdot \left( \frac{\log(1/\delta)}{\varepsilon} + \frac{\sqrt{dk\log(1/\delta)}}{\left(\frac{c_3\lambda}{1500\eta}\right) \cdot \varepsilon} \right),$$

where $c'$ denotes the constant from Fact B.26. Therefore we obtain by Fact B.26 (FriendlyCore averaging) that

$$\Pr\left[ \left\|\mathbf{P} - \widetilde{\mathbf{P}}\right\|_F \leq \underbrace{\frac{c_3\lambda\sqrt{k}\gamma}{12}}_{\alpha} \right] \geq 0.99,$$

where $\mathbf{P}$ is the $q \times d$ matrix whose rows are $\Pi\mathbf{p}_1, \ldots, \Pi\mathbf{p}_q$. By Fact B.15 we have that

$$\Pr\left[ \sigma_k(\mathbf{P}) \geq c_3\sqrt{k} \right] \geq 0.95,$$

and in the following we assume that the above event occurs (which in particular implies that $Span(\mathbf{P}^T) = Span(\Pi)$). Finally, since $2\alpha \leq \sigma_k(\mathbf{P})$ by assumption (and assuming $\gamma_{\max} \leq \frac{6}{c_3\lambda}$), we conclude by Proposition B.13 that

$$\left\|\Pi - \widetilde{\Pi}\right\|_F \leq 2\sqrt{2} \cdot \frac{\alpha}{\sigma_k(P) - \alpha} \leq \lambda\gamma/2.$$

We therefore conclude the proof of the lemma by Proposition B.16.

$\square$

## C.3 Strong Estimator

In this section, we prove Theorem 1.7, stated below.

**Theorem C.8** (Restatement of Theorem 1.7). *Let* $n, k, d \in \mathbb{N}$, $\lambda > 0$, *such that* $k \leq \min\{n, d\}$. *There exists an* $(k, \lambda, \beta = 0.8, \gamma_{\max})$*-weak subspace estimator* $\mathsf{M} \colon (\mathcal{S}_d)^n \times [0, 1] \to \mathbb{R}^{d \times d}$ *with*

$$\gamma_{\max} = \Omega\left( \min\{\frac{1}{\lambda}, \frac{\lambda^2\varepsilon^2}{\lambda^2\varepsilon\log(1/\delta) + \left(k + \log\left(\frac{dk\log(1/\delta)}{\lambda\varepsilon}\right)\right)dk\log(1/\delta)}\} \right)$$

*and*

$$n = O\left( k\log k \left( \frac{\log(1/\delta)}{\varepsilon} + \frac{\left(k + \log\left(\frac{dk\log(1/\delta)}{\lambda\varepsilon}\right)\right)dk\log(1/\delta)}{\lambda^2\varepsilon^2} \right) \right)$$

*such that* $\mathsf{M}(\cdot, \gamma)$ *is* $(\varepsilon, \delta)$*-DP for every* $\gamma \in [0, 1]$.

**Lemma C.9.** *Let* $c_1$, $c_2$, $c_3$ *be the constants from Facts B.14 to B.15 (respectively), and let* $c$ *be a large enough constant. Let* $t = c \cdot \left( \frac{\log(1/\delta)}{\varepsilon} + \frac{\left(k + \log\left(\frac{dk\log(1/\delta)}{\lambda\varepsilon}\right)\right)dk\log(1/\delta)}{\lambda^2\varepsilon^2} \right)$, $\eta = c_1 \cdot \left( \sqrt{k} + \sqrt{\ln(qt)} \right)$ *and* $q = c_2 \cdot k$. *Then for any* $n \geq 800k\ln(25k) \cdot t$, *the mechanism* $\mathsf{M} \colon (\mathcal{S}_d)^n \times [0, 1] \to \mathcal{W}_{d,k}$ *defined by* $\mathsf{M}(X, \gamma) := \mathsf{EstSubspace}_{k,t}^{\mathsf{SS\_Agg}_{\varepsilon,\delta,k,q,\xi=8\sqrt{2tk}\eta\gamma}}(X)$ *is an* $(k, \lambda, \beta = 0.8, \gamma_{\max} = \min\{\frac{1}{2t}, \frac{6}{c_3\lambda}\})$*-strong-subspace-estimator.*

*Proof.* Fix $X \in (\mathcal{S}_d)^n$ with $\frac{\sigma_{k+1}(X)}{\sigma_k(X)} \leq \gamma \leq \gamma_{\max}$ and let $\Pi \in \mathcal{W}_{d,k}$ be the projection of the top-$k$ rows subspace of $X$. Consider a random execution of $\mathsf{M}(X)$. Let $\{\Pi_j\}_{j=1}^t$, $\{\mathbf{p}_i\}_{i=1}^q$, $\{\mathbf{y}^j\}_{j=1}^t$, $\mathbf{z}, \widetilde{\mathbf{P}}$

be (random variables of) the values of $\{\Pi_j\}_{j=1}^t, \{p_i\}_{i=1}^q, \{\boldsymbol{y}^j\}_{j=1}^t, z, \widetilde{P}$ in the execution, and let $\widetilde{\boldsymbol{\Pi}}$ be the output. By Lemma C.4(1) (recall that $\gamma_{\max} \leq \frac{1}{2t}$) and the union bound,

$$\forall j \in [t]: \quad \Pr\Big[\|\boldsymbol{\Pi} - \boldsymbol{\Pi}_j\| \leq 4\sqrt{2t}\gamma\Big] \geq 0.99. \tag{5}$$

Let $\mathbf{a}_j = \mathbb{1}\{\|\boldsymbol{\Pi} - \boldsymbol{\Pi}_j\| \leq 4\sqrt{2t}\gamma\}$ (indicator random variable) and let $\mathbf{a} = \sum_{j=1}^t \mathbf{a}_j$. As in Equation (4), the above yields that

$$\Pr[\mathbf{a} \geq 0.8t] \geq 0.95. \tag{6}$$

In the following we assume that the event $\mathbf{a} \geq 0.8t$ occurs. Let $\mathbf{J} = \{j \in [t]: \mathbf{a}_j = 1\}$. By Fact B.14 and the definition of $\eta$ we obtain that

$$\Pr\Big[\forall i \in [q], j \in \mathbf{J}: \|(\boldsymbol{\Pi} - \boldsymbol{\Pi}_j)\mathbf{p}_i\|_2 \leq 4\sqrt{2t}\eta\gamma\Big] \geq 0.95,$$

In the following we assume that the above event occurs. This yields that

$$\forall i, j \in \mathbf{J}: \|\mathbf{y}_i - \mathbf{y}_j\|_2 \leq 8\sqrt{2tk}\eta\gamma = \xi.$$

Furthermore, by the definition of $t$ and $\eta$, it holds that

$$t \geq c' \cdot \left( \frac{\log(1/\delta)}{\varepsilon} + \frac{\sqrt{dk\log(1/\delta)}}{\left(\frac{c_3\lambda}{150\eta\sqrt{t}}\right) \cdot \varepsilon} \right),$$

where $c'$ denotes the constant from Fact B.26. Therefore we obtain by Fact B.26 (FriendlyCore averaging) that

$$\Pr\left[ \left\|\mathbf{P} - \widetilde{\mathbf{P}}\right\|_F \leq \underbrace{\frac{c_3\lambda\sqrt{k}\gamma}{12}}_{\alpha} \right] \geq 0.99,$$

where $\mathbf{P}$ is the $q \times d$ matrix whose rows are $\Pi\mathbf{p}_1, \dots, \Pi\mathbf{p}_q$. By Fact B.15 we have that

$$\Pr\Big[\sigma_k(\mathbf{P}) \geq c_3\sqrt{k}\Big] \geq 0.95,$$

and in the following we assume that the above event occurs (which in particular implies that $Span(\mathbf{P}^T) = Span(\Pi)$). Finally, since $2\alpha \leq \sigma_k(\mathbf{P})$ by assumption (and using $\gamma_{\max} = \frac{6}{c_3\lambda}$), we conclude by Proposition B.13 that

$$\left\|\boldsymbol{\Pi} - \widetilde{\boldsymbol{\Pi}}\right\|_F \leq 2\sqrt{2} \cdot \frac{\alpha}{\sigma_k(P) - \alpha} \leq \lambda\gamma/2.$$

We therefore conclude the proof of the lemma by Proposition B.16. $\qquad\square$

# D   Lower Bounds

In this section, we prove our lower bounds. In Appendix D.1 we prove Theorem 1.8 (lower bound for weak estimators) and in Appendix D.2 we prove Theorem 1.9 (lower bound for strong estimators). Both lower bounds rely on the framework of Peter et al. [2024], described in Appendix B.8.5.

Throughout this section, recall that for $d, k \in \mathbb{N}$ we denote by $\mathcal{W}_{d,k}$ the set of all $d \times d$ projection matrices of rank $k$, and denote by $\mathcal{P}_d$ the set of all $d \times d$ permutation matrices.

## D.1   Weak Estimators

**Theorem D.1** (Restatement of Theorem 1.8). *Let $n, k, d \in \mathbb{N}$, $\lambda \geq 1$, $\beta \in (0,1]$ such that $d \geq ck$ and $\lambda^2 \leq \frac{d}{ck\log k}$ for large enough constant $c > 0$, and $n$ is a multiple of $k$. If $\mathsf{M}: (\mathcal{S}_d)^n \times [0,1] \to \mathcal{W}_{d,k}$ is a $(k, \lambda, \beta, \gamma_{\max} = \frac{1}{10^6\lambda^2})$-weak subspace estimator and $\mathsf{M}(\cdot, \gamma)$ is $\left(1, \frac{\beta}{5nk}\right)$-DP for every $\gamma \in [0,1]$, then $n \geq \Omega\left(\frac{\sqrt{kd}/\lambda}{\log^{1.5}\left(\frac{dk}{\lambda\beta}\right)}\right)$.*

Theorem 1.8 is an immediate corollary of Lemma B.34 (Framework for lower bounds) and the following Lemma D.2.

**Lemma D.2.** *Let $n, k, d, \lambda, \beta$ and $\mathsf{M}$ as in Theorem D.1. Let $\alpha = \frac{1}{5 \cdot 10^6 \lambda^2 k}$, $n_0 = n/k$, $\ell = 2 \cdot \lceil \frac{1}{4}(1-\alpha)d \rceil$, $d_0 = d - 2\ell$, $\mathcal{X} = \mathcal{S}_d$, $\mathcal{Z} = [k] \times (\mathcal{P}_d)^k$ and $\mathcal{W} = \mathcal{W}_{d,k}$. Let $\mathsf{G} \colon \{-1,1\}^{n_0 \times d_0} \times \mathcal{Z} \to \mathcal{X}^n$ be Algorithm D.3, and let $\mathsf{F} \colon \mathcal{Z} \times \mathcal{W} \to [-1,1]^{d_0}$ be Algorithm D.4. Then the triplet $(\mathsf{M}, \mathsf{F}, \mathsf{G})$ is $\frac{0.8\beta}{k}$-leaking (Definition B.33).*

Note that by Lemmas D.2 and B.34, we obtain that $n_0 \geq \Omega\left(\frac{\sqrt{d_0}}{\log^{1.5}(d_0/\beta)}\right)$, but since $n_0 = n/k$ and $d_0 = \Theta(\alpha d) = \Theta\left(\frac{d}{\lambda^2 k}\right)$, the proof of Theorem D.1 follows. We next prove Lemma D.2.

---

**Algorithm D.3** (Algorithm $\mathsf{G}$).

*Parameters: $n_0, n, d_0, d, \ell \in \mathbb{N}$.*

*Inputs: $z = (s, (P_1, \ldots, P_k))$ for $s \in [k]$ and $P_1, \ldots, P_k \in \mathcal{P}_d$, and a matrix $X \in \{-1,1\}^{n_0 \times d_0}$.*

*Operation:*

  1. *Sample $A = (A_1, \ldots, A_k) \sim \mathcal{D}(n_0, d_0)^k$, and set $A_s = X$.*
  2. *For $t \in [k]$, compute $B_t = \mathsf{PAP}_{n_0, d_0, \ell}(A_t, P_t) \in \{-1,1\}^{n_0 \times d}$ (Definition B.35), and let $B \in \{-1,1\}^{n \times d}$ be the vertical concatenation of $B_1, \ldots, B_k$.*
  3. *Output $Y = \frac{1}{\sqrt{d}} B \in (\mathcal{S}_d)^n$.*

---

**Algorithm D.4** (Algorithm $\mathsf{F}$).

*Parameters: $n_0, n, d_0, d, \ell \in \mathbb{N}$.*

*Inputs: $z = (s, (P_1, \ldots, P_k))$ for $s \in [k]$ and $P_1, \ldots, P_k \in \mathcal{P}_d$, and a rank-$k$ projection matrix $\widetilde{\Pi} \in \mathcal{W}$ (which is the output of $\mathsf{M}(Y, \gamma = \frac{1}{1000\lambda})$ ).*

*Operation:*

  1. *Compute a vector $u = (u^1, \ldots, u^d) \in Span(\widetilde{\Pi} \cdot P_s^T)$ that maximizes $\min\{\sum_{j=d_0+1}^{d_0+\ell/2} \mathsf{sign}(u^j), -\sum_{j=d_0+\ell+1}^{d_0+3\ell/2} \mathsf{sign}(u^j)\}$.*
  2. *Output $q = \mathsf{sign}(u)^{1,\ldots,d_0} \in \{-1,1\}^{d_0}$.*

---

### D.1.1 Proving Lemma D.2

In the following, we define random variables $\mathbf{X} \sim \mathcal{D}(n_0, d_0)$ (Definition B.30) and $\mathbf{z} = (\mathbf{s}, (\mathbf{P}_1, \ldots, \mathbf{P}_k)) \leftarrow \mathcal{Z}$, and consider a random execution of $\mathsf{A}^{\mathsf{M},\mathsf{F},\mathsf{G}}(\mathbf{X}) = \mathsf{F}(\mathbf{z}, \mathsf{M}(\mathsf{G}(\mathbf{X}, \mathbf{z})))$. Let $\mathbf{B}_1, \ldots, \mathbf{B}_k, \mathbf{B}, \mathbf{Y}$ be the values of $B_1, \ldots, B_k \in \{-1,1\}^{n_0 \times d}$, $B \in \{-1,1\}^{n \times d}$ and $Y \in (\mathcal{S}_d)^n$ in the execution of $\mathsf{G}$, and let $\mathbf{Y}_1 = \frac{1}{\sqrt{d}} \mathbf{B}_1, \ldots, \mathbf{Y}_k = \frac{1}{\sqrt{d}} \mathbf{B}_k$ (note that $\mathbf{Y}$ is a vertical concatenation of $\mathbf{Y}_1, \ldots, \mathbf{Y}_k$). Let $\mathbf{u}$ be the value of $u \in \mathcal{S}_d$ in the execution of $\mathsf{F}$. For $b \in \{-1,1\}$ let $\mathbf{F}_b$ be the set of $b$-marked columns of $\mathbf{B}_s \cdot \mathbf{P}_s^T$ (note that $\mathbf{F}_1$ includes $d_0 + 1, \ldots, d_0 + \ell$ and $\mathbf{F}_{-1}$ includes $d_0 + \ell + 1, \ldots, d$). Let $\mathcal{H}_1 = \{d_0 + 1, \ldots, d_0 + \ell/2\} \subseteq \mathbf{F}_1$ and $\mathcal{H}_{-1} = \{d_0 + \ell + 1, \ldots, d_0 + 3\ell/2\} \subseteq \mathbf{F}_{-1}$, and let $\mathcal{H} = \mathcal{H}_1 \cup \mathcal{H}_{-1}$. For $t \in [k]$, define

$$\mathbf{v}_t = \frac{1}{\sqrt{2\ell}} \cdot (\underbrace{0\ldots,0}_{d_0}, \underbrace{1,\ldots\ldots\ldots,1}_{\ell}, \underbrace{-1,\ldots\ldots\ldots,-1}_{\ell}) \cdot \mathbf{P}_t \in \mathcal{S}_d \tag{7}$$

The following claim holds under our assumption that $\lambda^2 \leq \frac{d}{ck \log k}$ for large enough constant $c$.

**Claim D.5.** *Let $\gamma = \frac{1}{1000\lambda}$. It holds that*

$$\Pr\left[\sum_{i=k+1}^n \sigma_i^2(\mathbf{Y}) \leq \gamma^2 \cdot \sigma_k^2(\mathbf{Y})\right] \geq 0.9.$$

*Proof.* Recall that $d = d_0 + 2\ell$ and $\ell \geq \frac{1}{2}(1 - \alpha)d$ for $\alpha = \frac{1}{5 \cdot 10^6 \lambda^2 k}$. First, note that

$$\Pr[\mathbf{v}_1, \ldots, \mathbf{v}_k \text{ are linearly independent}] \geq 1 - \left(2^{-2\ell} + 2^{-2\ell+1} + \ldots + 2^{-2\ell+(k-2)}\right) \geq 0.99, \tag{8}$$

where the last inequality holds since $\ell \approx d/2 \geq c \cdot k/2$ for large enough constant $c > 0$. Furthermore, note that for every $s, t \in [k]$, $2\ell \cdot \langle \mathbf{v}_s, \mathbf{v}_t \rangle = \sum_{j: \, \text{sign}(\mathbf{v}_s^j)=1} \text{sign}(\mathbf{v}_t^j) - \sum_{j: \, \text{sign}(\mathbf{v}_s^j)=-1} \text{sign}(\mathbf{v}_t^j)$ where each sum has Hypergeometric distributions $\mathcal{HG}_{2\ell,0,\ell}$ (Definition B.5). Therefore by Fact B.6 and the union bound, it holds that

$$\Pr\left[\forall s, t \in [k] : |\langle \mathbf{v}_s, \mathbf{v}_t \rangle| \leq O\left(\sqrt{\frac{\log k}{d}}\right)\right] \geq 0.99. \tag{9}$$

(i.e., $\mathbf{v}_1, \ldots, \mathbf{v}_k$ are almost orthogonal).

In the following, we assume that the events in Equations (8) and (9) occur. Using the Gram–Schmidt process on $\mathbf{v}_1, \ldots, \mathbf{v}_k$, we obtain orthogonal basis $\mathbf{u}_1, \ldots, \mathbf{u}_k$ to $Span\{\mathbf{v}_1, \ldots, \mathbf{v}_k\}$ such that for every $t \in [k]$, $\mathbf{u}_t = \mathbf{v}_t + \lambda_{t-1}\mathbf{v}_{t-1} + \mathbf{w}_t$, where $|\lambda_{t-1}| \leq O\left(\sqrt{\frac{\log k}{d}}\right)$ and $\|\mathbf{w}_t\|_2 \leq O\left(\frac{\log k}{d}\right)$ (holds by Proposition B.17). Recall that $\mathbf{Y}$ is a vertical concatenation of $\frac{1}{\sqrt{d}}\mathbf{B}_1, \ldots, \frac{1}{\sqrt{d}}\mathbf{B}_k$ and for every $t \in [k]$, the rows of $\mathbf{B}_t$ are all in

$$\{-1, 1\}^{d_0} \times (\underbrace{1, \ldots, 1}_{\ell}, \underbrace{-1, \ldots, -1}_{\ell}) \cdot \mathbf{P}_t = \{-1, 1\}^{d_0} \times (\underbrace{0, \ldots \ldots, 0}_{2\ell}) \cdot \mathbf{P}_t + \sqrt{\frac{2\ell}{d}} \cdot \mathbf{v}_t$$

Therefore, we obtain that

$$\|\mathbf{Y} \cdot \mathbf{u}_t\|_2^2 \geq \|\mathbf{Y}_t \cdot \mathbf{u}_t\|_2^2 \tag{10}$$

$$\geq \left(\langle\sqrt{\frac{2\ell}{d}}\mathbf{v}_t, \mathbf{u}_t\rangle^2 - \frac{d_0}{d}\right) \cdot n_0$$

$$\geq \left((1 - \alpha) \cdot \left(1 - O\left(\frac{\log k}{d}\right)\right)^2 - \alpha\right) \cdot n_0$$

$$\geq (1 - 4\alpha) \cdot n_0,$$

where the last inequality holds whenever $\alpha \geq \Theta(\log k/d)$, which holds by the assumption on $\lambda$. We therefore obtain that $\sigma_1^2(\mathbf{Y}), \ldots, \sigma_k^2(\mathbf{Y}) \geq (1 - 4\alpha) \cdot n_0$ which yields that $\sum_{i=k+1}^n \sigma_i^2(\mathbf{Y}) \leq n - k(1 - 4\alpha) \cdot n_0 = 4\alpha n$. Hence

$$\frac{\sum_{i=k+1}^n \sigma_i^2(\mathbf{Y})}{\sigma_k^2(\mathbf{Y})} \leq \frac{4\alpha n}{(1 - 4\alpha) \cdot \frac{n}{k}} = \frac{4\alpha k}{1 - 4\alpha} \leq \frac{1}{10^6 \lambda^2} = \gamma^2, \tag{11}$$

where the second inequality holds since $\alpha = \frac{1}{5 \cdot 10^6 \lambda^2 k}$.

$\square$

The following claim holds under our assumption that $d \geq ck$ for large enough constant $c > 0$.

**Claim D.6.** *It holds that*

$$\Pr\left[\text{sign}(\mathbf{u})^{[d]\backslash\mathcal{H}} \text{ strongly-agrees with } \left(\mathbf{B}_s\mathbf{P}_s^T\right)^{[d]\backslash\mathcal{H}}\right] \geq \frac{0.8\beta}{k}.$$

*where "strongly-agrees" is according to Definition B.36.*

*Proof.* In the following we assume that the 0.9 probability event in Claim D.5 occurs. Since M is $(k, \lambda, \beta, \gamma_{\max} = \frac{1}{1000\lambda})$-subspace estimator, it follows from Equation (11) that w.p. $\beta$, the output $\widetilde{\mathbf{\Pi}}$ of M($\mathbf{Y}$) satisfy

$$\left\|\widetilde{\mathbf{\Pi}} \cdot \mathbf{Y}^T\right\|_F^2 \geq \left\|\mathbf{\Pi} \cdot \mathbf{Y}^T\right\|_F^2 - \frac{n}{1000}, \tag{12}$$

where we denote by $\mathbf{\Pi}$ the projection matrix onto $Span\{\mathbf{v}_1, \ldots, \mathbf{v}_k\}$ (defined in Equation (7)). In the following, we assume that the event in (12) occurs. This yields that there must exists $\mathbf{t} \in [k]$ such that

$$
\begin{aligned}
\left\|\widetilde{\mathbf{\Pi}} \cdot \mathbf{Y}_{\mathbf{t}}^T\right\|_F^2 &\geq \left\|\mathbf{\Pi} \cdot \mathbf{Y}_{\mathbf{t}}^T\right\|_F^2 - \frac{n}{1000k} \\
&\geq \frac{2\ell}{d} \cdot \|\mathbf{v_t}\|_2^2 \cdot \frac{n}{k} - \frac{n}{1000k} \\
&\geq (0.999 - \alpha) \cdot \frac{n}{k}
\end{aligned}
$$

Since $\mathbf{s}$ (part of $\mathbf{z}$) is chosen at random and does not change the distribution of $\mathbf{Y}$ (the input of the mechanism), w.p. $1/k$ the above holds for $\mathbf{t} = \mathbf{s}$, i.e.,

$$
\left\|\widetilde{\mathbf{\Pi}} \cdot \mathbf{Y}_{\mathbf{s}}^T\right\|_F^2 \geq (0.999 - \alpha) \cdot \frac{n}{k}. \tag{13}
$$

In the following we assume that the $1/k$-probability event in Equation (13) occurs.

Recall that $\mathbf{B}_{\mathbf{s}} = \sqrt{d} \cdot \mathbf{Y}_{\mathbf{s}}$, and let $\widetilde{\mathbf{\Pi}}_{\mathbf{s}} = \widetilde{\mathbf{\Pi}} \cdot \mathbf{P}_{\mathbf{s}}^T$ and $\mathbf{B}_{\mathbf{s}}' = \mathbf{B}_{\mathbf{s}} \cdot \mathbf{P}_{\mathbf{s}}^T$. It follows that

$$
\left\|\widetilde{\mathbf{\Pi}}_{\mathbf{s}} \cdot (\mathbf{B}_{\mathbf{s}}')^T\right\|_F^2 = d \cdot \left\|\widetilde{\mathbf{\Pi}} \cdot \mathbf{Y}_{\mathbf{s}}^T\right\|_F^2 \geq (0.999 - \alpha) \cdot \frac{dn}{k} \tag{14}
$$

In the following, define

$$
\mathbf{v} = (\mathbf{v}^1, \ldots, \mathbf{v}^d) \in \{-1, 0, 1\}^d \quad \text{where} \quad \mathbf{v}^j = \begin{cases} 1 & j \in \mathbf{F}_1 \\ -1 & j \in \mathbf{F}_{-1} \\ 0 & \text{o.w.} \end{cases} \tag{15}
$$

Note that each row $i$ of $\mathbf{B}_{\mathbf{s}}'$ can be written as $\mathbf{v} + \xi_i$ where $\xi_i \in \{-1, 0, 1\}^{d_0} \times \underbrace{(0, \ldots, 0)}_{2\ell}$. This yields that

$$
\begin{aligned}
\left\|\widetilde{\mathbf{\Pi}}_{\mathbf{s}} \cdot \mathbf{v}^T\right\|_2^2 &= \frac{k}{n} \cdot \left\|\widetilde{\mathbf{\Pi}}_{\mathbf{s}} \cdot (\mathbf{B}_{\mathbf{s}}')^T\right\|_F^2 - \frac{k}{n} \sum_{i=1}^{n/k} \left\|\widetilde{\mathbf{\Pi}}_{\mathbf{s}} \cdot \xi_i^T\right\|_2^2 \\
&\geq (0.999 - \alpha)d - d_0 \geq (0.999 - 2\alpha)d
\end{aligned}
$$

Now, since

$$
\|\mathbf{v}\|_2^2 = \langle \mathbf{v}, \mathbf{v} \rangle = \langle \widetilde{\mathbf{\Pi}}_t \mathbf{v}^T + (I - \widetilde{\mathbf{\Pi}}_t)\mathbf{v}^T, \widetilde{\mathbf{\Pi}}_t \mathbf{v}^T + (I - \widetilde{\mathbf{\Pi}}_t)\mathbf{v}^T \rangle = \left\|\widetilde{\mathbf{\Pi}}_t \mathbf{v}^T\right\|_2^2 + \left\|(I - \widetilde{\mathbf{\Pi}}_t)\mathbf{v}^T\right\|_2^2,
$$

we conclude that for $\tilde{\mathbf{v}}^T = \widetilde{\mathbf{\Pi}}_t \mathbf{v}^T \in Span(\widetilde{\mathbf{\Pi}}_t)$:

$$
\|\mathbf{v} - \tilde{\mathbf{v}}\|_2^2 = \|\mathbf{v}\|_2^2 - \left\|\widetilde{\mathbf{\Pi}}_t \mathbf{v}^T\right\|_2^2 \leq d - (0.999 - 2\alpha)d \leq \frac{d}{500}. \tag{16}
$$

We next define $(\mathcal{I}, \eta)$-*good* vectors.

**Definition D.7.** *We say that a vector* $\mathbf{w} \in \{-1, 1\}^d$ *is* $(\mathcal{I}, \eta)$-*good iff for* $(1 - \eta)$ *fraction of the indices* $j \in \mathcal{I}$ *it holds that* $\mathsf{sign}(\mathbf{w}^j) = \mathsf{sign}(\mathbf{v}^j)$ *(for the* $\mathbf{v}$ *defined in Equation (15)).*

We use the following trivial fact:

**Observation D.8.** *If* $\mathbf{w}$ *is not* $(\mathcal{I}, \eta)$-*good, then* $\|\mathbf{w} - \mathbf{v}\|_2^2 \geq \eta|\mathcal{I}|$.

Since for both $b \in \{-1, 1\}$, $|\mathcal{H}_b| \geq \frac{1}{4}(1-\alpha)d \geq d/5$ and $\|\mathbf{v} - \tilde{\mathbf{v}}\|_2^2 \leq \frac{d}{500}$, Observation D.8 implies that $\tilde{\mathbf{v}}$ is $(\mathcal{H}_1, \eta)$-good and $(\mathcal{H}_{-1}, \eta)$-good for $\eta = \frac{1}{100}$. Therefore, the vector $\mathbf{u}$ (computed in F) is also $(\mathcal{H}_1, \eta)$-good and $(\mathcal{H}_{-1}, \eta)$-good.

In the following, we use similar arguments to Dwork et al. [2014] for claiming that because $\mathbf{u}$ is good on half of the padding location, then it should also be good on the rest of the marked locations.

From the point of view of the algorithm M (which does not know $\mathbf{P_s}$), the locations in $\mathcal{H}_b$ are indistinguishable from those in $\mathbf{F}_b \setminus \mathcal{H}_b$. Therefore, for any point that is not $(\mathbf{F}_b, 3\eta)$-good, the probability (taken over the random choice of $\mathbf{P_s}$) that it is $(\mathcal{H}_b, 2\eta)$-good is at most $\exp(-\Omega(\eta^2 d))$. Now let $\mathbf{N}$ be an $(\eta\sqrt{d})$-net of the $\sqrt{d}$-sphere in $Span(\widetilde{\mathbf{\Pi}}_\mathbf{s})$ (Definition B.9). Taking a union bound over a $\exp(O(k \log(1/\eta))) = \exp(O(k))$ points in $\mathbf{N}$ (Fact B.10), and recall that $d \geq ck$ for large enough constant $c$, we conclude that except with probability $\exp(O(k)) \cdot \exp(-\Omega(\eta^2 d)) \leq 0.01$, any given vector in $\mathbf{N}$ that is $(\mathcal{H}_b, 2\eta)$-good is also $(\mathbf{F}_b \setminus \mathcal{H}_b, 4\eta)$-good. Since $\mathbf{u}$ is $(\mathcal{H}_b, \eta)$-good, then its nearest net point $\mathbf{u}'$ is $(\mathcal{H}_b, 2\eta)$-good. Thus $\mathbf{u}'$ is $(\mathbf{F}_b \setminus \mathcal{H}_b, 4\eta)$-good which implies that $\mathbf{u}$ is $(\mathbf{F}_b \setminus \mathcal{H}_b, 5\eta)$-good except w.p. $\exp(-\Omega(\eta^2 d)) \leq 0.01$. But by definition of $\mathbf{v}$, it perfectly agrees with the marked columns of $\mathbf{B_s} \cdot \mathbf{P_s}^T$. Since $5\gamma < 0.1$ the above implies that $sign(\mathbf{u})^{\mathbf{F}_b \setminus \mathcal{H}_b}$ strongly-agrees (Definition B.36) with the matrix $\left(\mathbf{B_s} \cdot \mathbf{P_s}^T\right)^{\mathbf{F}_b \setminus \mathcal{H}_b}$ which implies that $sign(\mathbf{u})^{[d] \setminus \mathcal{H}_b}$ strongly-agrees with the matrix $\left(\mathbf{B_s} \cdot \mathbf{P_s}^T\right)^{[d] \setminus \mathcal{H}_b}$, as required.

$\square$

We now ready to prove the final claim that concludes the proof of Lemma D.2.

**Claim D.9.** *It holds that*

$$\Pr_{r,r' \leftarrow \{0,1\}^m, \, X \sim \mathcal{D}}\left[\mathsf{A}^{\mathsf{M}_r, \mathsf{F}, \mathsf{G}_{r'}}(X) \text{ is strongly-correlated with } X\right] \geq \frac{0.8\beta}{k}.$$

*Proof.* In the following we assume that the event from the statement of Claim D.6 occurs, and let $\mathcal{H} = \mathcal{H}_1 \cup \mathcal{H}_{-1}$. Define the permutation matrix $\mathbf{P}' \in \mathcal{P}_{d_0 + \ell}$ that is obtained by removing the rows $\mathcal{H}$ and the columns $\mathbf{P_s}(\mathcal{H})$ from $\mathbf{P_s}$ (i.e., $\mathbf{P}'$ is the permutation induced by $\mathbf{P_s}$ between $[d] \setminus \mathcal{H}$ and $[d] \setminus \mathbf{P_s}(\mathcal{H})$). Similarly, define the permutation matrix $\overline{\mathbf{P}}' \in \mathcal{P}_\ell$ that is obtained by removing the rows $\overline{\mathcal{H}} = [d] \setminus \mathcal{H}$ and the columns $\mathbf{P_s}(\overline{\mathcal{H}})$ from $\mathbf{P_s}$ (i.e., $\overline{\mathbf{P}}'$ is the permutation induced by $\mathbf{P_s}$ between $\mathcal{H}$ and $\mathbf{P_s}(\mathcal{H})$). Note that $\mathbf{P}'$ is distributed uniformly over $\mathcal{P}_{d_0 + \ell}$ for any choice of $\overline{\mathbf{P}}'$. In the following, let $\widetilde{\mathbf{\Pi}}' = \widetilde{\mathbf{\Pi}}^{[d] \setminus \mathcal{H}}$, $\mathbf{q}' = sign(\mathbf{u})^{[d] \setminus \mathcal{H}} \cdot \mathbf{P}'$, and $\mathbf{B}' = \mathbf{B_s}^{[d] \setminus \mathcal{H}}$. By Claim D.6 it holds that

$$\Pr[\mathbf{q}' \text{ strongly-agrees with } \mathbf{B}'] \geq \frac{0.8\beta}{k}. \tag{17}$$

But note that $\mathbf{B}' = \mathsf{PAP}_{n_0, d_0, \ell/2}(\mathbf{X}, \mathbf{P}')$ and also note that $\mathbf{q}'$ is just a function of $\widetilde{\mathbf{\Pi}}'$ and $\overline{\mathbf{P}}'$ (i.e., independent of $\mathbf{P}'$) since it equals to $sign(\mathbf{w})^{\overline{\mathbf{P}}'([d] \setminus \mathcal{H})}$ where $\mathbf{w}$ is the vector in $Span(\widetilde{\mathbf{\Pi}})$ that maximizes $\min\{\sum_{j \in \overline{\mathbf{P}}'(\mathcal{H}_1)} sign(\mathbf{w}^j), -\sum_{j \in \overline{\mathbf{P}}'(\mathcal{H}_{-1})} sign(\mathbf{w}^j)\}$. Furthermore, note that $(\mathbf{q}' \cdot (\mathbf{P}')^T)^{1,\dots,d_0} = \mathbf{q}$, where $\mathbf{q}$ is the final output of $\mathsf{A}^{\mathsf{M}, \mathsf{F}, \mathsf{G}}(\mathbf{X}) = \mathsf{F}(\mathbf{z}, \mathsf{M}(\mathsf{G}(\mathbf{X}, \mathbf{z})))$. Thus by Lemma B.37 and Equation (17) we conclude that

$$\Pr_{r,r' \leftarrow \{0,1\}^m, \, X \sim \mathcal{D}}\left[\mathsf{A}^{\mathsf{M}_r, \mathsf{F}, \mathsf{G}_{r'}}(X) \text{ is strongly-correlated with } X\right] \geq \frac{0.8\beta}{k}.$$

$\square$

## D.2 Strong Estimators

**Theorem D.10** (Restatement of Theorem 1.9). *Let $n, k, d \in \mathbb{N}$, $\lambda \geq 1$, $\beta \in (0,1]$ such that $d \geq ck$ and $\lambda^2 \leq \frac{d}{c \log k}$ for large enough constant $c > 0$, and $n$ is a multiple of $k$. If $\mathsf{M} \colon (\mathcal{S}_d)^n \times [0,1] \to \mathbb{R}^{d \times d}$ is an $(k, \lambda, \beta, \gamma_{\max} = \frac{1}{10^6 \lambda^2})$-strong subspace estimator and $\mathsf{M}(\cdot, \gamma)$ is $\left(1, \frac{\beta}{5nk}\right)$-DP for every $\gamma \in [0,1]$, then $n \geq \Omega\left(\frac{k\sqrt{d}/\lambda}{\log^{1.5}\left(\frac{dk}{\lambda\beta}\right)\sqrt{\log(dk)\log(2n/k)}}\right).$*

We prove Theorem D.10 using a similar technical lemma to Lemma D.2, but now since $\mathsf{M}$ is a strong subspace estimator, we can use $\alpha = \tilde{\Theta}\left(\frac{1}{\lambda^2}\right)$ (rather than $\Theta\left(\frac{1}{\lambda^2 k}\right)$ as in Lemma D.2).

**Lemma D.11.** *There exists large enough constant $c > 0$ such that the following holds: Let $n, k, d, \lambda, \beta$ and $\mathsf{M}$ as in Theorem D.10, let $\alpha = \frac{1}{c \cdot \log(dk) \log(2n/k) \cdot \lambda^2}$, $n_0 = n/k$, $\ell = 2 \cdot \left\lceil \frac{1}{4}(1 - \alpha)d \right\rceil$, $d_0 = d - 2\ell$, $\mathcal{X} = \mathcal{S}_d$, $\mathcal{Z} = [k] \times (\mathcal{P}_d)^k$ and $\mathcal{W} = \mathcal{W}_{d,k}$. Let $\mathsf{G} \colon \{-1,1\}^{n_0 \times d_0} \times \mathcal{Z} \to \mathcal{X}^n$ be Algorithm D.3, and let $\mathsf{F} \colon \mathcal{Z} \times \mathcal{W} \to [-1,1]^{d_0}$ be Algorithm D.4. Then the triplet $(\mathsf{M}, \mathsf{F}, \mathsf{G})$ is $\frac{0.8\beta}{k}$-leaking (Definition B.33).*

By Lemmas D.11 and B.34, it holds that $n_0 \geq \Omega\left(\frac{\sqrt{d_0}}{\log^{1.5}(d_0/\beta)}\right)$. The proof of Theorem D.10 now follows since $n_0 = n/k$ and $d_0 \approx \alpha d$ for the $\alpha$ defined in the lemma.

### D.2.1 Proving Lemma D.11

As in the proof of Lemma D.2, we define random variables $\mathbf{X} \sim \mathcal{D}(n_0, d_0)$ (Definition B.30) and $\mathbf{z} = (\mathbf{s}, (\mathbf{P}_1, \ldots, \mathbf{P}_k)) \leftarrow \mathcal{Z}$, and consider a random execution of $\mathsf{A}^{\mathsf{M},\mathsf{F},\mathsf{G}}(\mathbf{X}) = \mathsf{F}(\mathbf{z}, \mathsf{M}(\mathsf{G}(\mathbf{X}, \mathbf{z})))$. Let $\{\mathbf{A}_t\}, \{\mathbf{B}_t\}, \mathbf{Y}$ be the values of $\{A_t\}, \{B_t\}, Y \in (\mathcal{S}_d)^n$ in the execution of $\mathsf{G}$, and recall that $\mathbf{Y}$ is a vertical concatenation of $\mathbf{Y}_1, \ldots, \mathbf{Y}_k$ where $\mathbf{Y}_t = \frac{1}{\sqrt{d}}\mathbf{B}_t$.

The only difference from proving Lemma D.2 is to prove a different version of Claim D.5 that only considers the gap between $\sigma_k$ and $\sigma_{k+1}$ (which will meet the requirements of the strong estimator $\mathsf{M}$). Namely, is suffices to prove the following claim:

**Claim D.12.** *It holds that*

$$\Pr[\sigma_{k+1}(\mathbf{Y}) \leq \gamma \cdot \sigma_k(\mathbf{Y})] \geq 0.9,$$

*for $\gamma = \frac{1}{1000\lambda}$.*

*Proof of Claim D.12.* As in Equation (7), for $t \in [k]$ we define

$$\mathbf{v}_t = \frac{1}{\sqrt{2\ell}} \cdot (\underbrace{0 \ldots, 0}_{d_0}, \underbrace{1, \ldots \ldots \ldots, 1}_{\ell}, \underbrace{-1, \ldots \ldots \ldots, -1}_{\ell}) \cdot \mathbf{P}_t \ \in \ \mathcal{S}_d \tag{18}$$

Let $\mathbf{E} = Span\{\mathbf{v}_1, \ldots, \mathbf{v}_k\}$. Note that by construction, $\{\mathbf{v}_t\}, \mathbf{E}$ are independent of $\{\mathbf{A}_t\}$.

Similarly to Equation (9) it holds that

$$\Pr\left[\forall s, t \in [k] : |\langle \mathbf{v}_s, \mathbf{v}_t \rangle| \leq O\left(\sqrt{\frac{\log k}{d}}\right)\right] \geq 0.99. \tag{19}$$

This yields (using similar steps as in the proof of Claim D.5) that w.p. $0.98$, $\dim(\mathbf{E}) = k$ and every unit vector $\mathbf{u} \in \mathbf{E}$ has $\|\mathbf{Y}\mathbf{u}\|_2^2 \geq (1 - 4\alpha)\frac{n}{k}$. This in particular implies that $\sigma_k^2(\mathbf{Y}) \geq (1 - 4\alpha)\frac{n}{k}$.

Our goal is to prove that w.p. $0.99$ it also holds that $\sigma_{k+1}^2(\mathbf{Y}) \leq \tilde{O}(\alpha)\frac{n}{k}$. Let $\bar{\mathbf{E}}$ be the orthogonal subspace to $\mathbf{E}$. Our goal is reduced to showing that there exists a constant $c$ such that

$$\Pr\left[\forall u \in \bar{\mathbf{E}} \cap \mathcal{S}_d : \quad \|\mathbf{Y}u\|_2^2 \leq \alpha \cdot c \log(k/\alpha) \log(2n/k) \cdot \frac{n}{k}\right] \geq 0.99. \tag{20}$$

Given that the event in Equation (20) holds we conclude that $\sigma_{k+1}^2(\mathbf{Y}) \leq \alpha \cdot c \log(k/\alpha) \log(2n/k) \cdot \frac{n}{k}$ and hence

$$\frac{\sigma_{k+1}^2(\mathbf{Y})}{\sigma_k^2(\mathbf{Y})} \leq \frac{\alpha \cdot c \log(k/\alpha) \log(2n/k) \cdot \frac{n}{k}}{(1 - 4\alpha) \cdot \frac{n}{k}} \leq \frac{1}{10^6 \lambda^2} = \gamma^2,$$

where the last inequality holds by taking

$$\alpha = \frac{1}{2 \cdot 10^6 \cdot c \log(k/\alpha) \cdot \log(2n/k) \cdot \lambda^2}.$$

Note that for every $\mathbf{u} \in \bar{\mathbf{E}} \cap \mathcal{S}_d$ is holds that

$$\|\mathbf{Y}\mathbf{u}\|_2^2 = \sum_{t=1}^{k}\|\mathbf{Y}_t\mathbf{u}\|_2^2 = \frac{1}{d}\sum_{t=1}^{k}\|\mathbf{A}_t\mathbf{u}\|_2^2,$$

where the last equality holds since $\mathbf{u}$ is orthogonal to $\mathbf{v}_1,\ldots,\mathbf{v}_k$. Therefore, we can prove Equation (20) by proving that there exists a constant $c'$ such that

$$\forall u \in \mathcal{S}_d: \quad \Pr\left[\frac{1}{d}\cdot\sum_{t=1}^{k}\|\mathbf{A}_t u\|_2^2 > \alpha\cdot c' \log(k/\alpha)\log(2n/k)\cdot\frac{n}{k}\right] \le \exp(-d\ln(3k/\alpha)-10). \tag{21}$$

Given that Equation (21) holds, we prove the claim using a net argument. By taking an $\exp(d\ln(3k/\alpha))$-size $\sqrt{\alpha/k}$-net of $\mathcal{S}_d$ (Fact B.10), Equation (20) follows by Equation (21) and the union bound over all the net points, which concludes the proof of the claim.

In the following we focus on proving Equation (21). Fix a columns vector $u \in \mathcal{S}_d$. Recall that $\mathbf{A}_t \sim \mathcal{D}(n_0, d_0)$ (Definition B.30), and $\mathbf{A}_t$ is located in $d_0$ random columns out of $d$, which are the columns $\mathbf{J} = \mathbf{P}_t([d_0])$. Let $\mathbf{a}_{t,i} \in \{\pm\frac{1}{\sqrt{d}}\}^{d_0}$ be the $i^{\text{th}}$ row of $\mathbf{A}_t$. By Definition B.30, the coordinates of $\mathbf{a}_{t,i}$ are i.i.d. Bernoulli distribution over $\{-1, 1\}$, each takes 1 w.p. $1/2$ and $-1$ w.p. $1/2$. Therefore by Fact B.2 it holds that

$$\forall\xi \ge 0: \quad \Pr\left[\langle \mathbf{a}_{i,t}, u\rangle^2 \ge \xi\right] = \mathrm{E}_\mathbf{J}\left[\Pr\left[\langle \mathbf{a}_{i,t}, u_\mathbf{J}\rangle^2 \ge \xi\right]\right] \le \mathrm{E}_\mathbf{J}\left[2\exp\left(-\frac{d\xi}{2\|u_\mathbf{J}\|_2^2}\right)\right]$$

$$\le 2\exp\left(-\frac{d\xi}{\mathrm{E}\left[2\|u_\mathbf{J}\|_2^2\right]}\right) \le 2\exp\left(-\frac{d\xi}{2\alpha}\right),$$

where the first inequality holds by Fact B.2, the second one holds by Jensen's inequality (and since the function $f(x) = e^{-1/x}$ is concave), and the last one hold since $\mathrm{E}\left[\|u_\mathbf{J}\|_2^2\right] = \frac{d_0}{d} \le \alpha$. By the union bound over the $n/k$ rows of $\mathbf{A}_t$ we obtain that

$$\forall\xi \ge 0: \quad \Pr\left[\|\mathbf{A}_t w\|_2^2 \ge \frac{\xi n}{k}\right] \le \frac{2n}{k}\cdot\exp\left(-\frac{d\xi}{2\alpha}\right),$$

or equivalently

$$\forall\xi \ge 0: \quad \Pr\left[\|\mathbf{Y}_t u\|_2^2 \ge \xi\right] = \Pr\left[\|\mathbf{A}_t w\|_2^2 \ge \xi\right] \le \frac{2n}{k}\cdot\exp\left(-\frac{dk}{2\alpha n}\cdot\xi\right). \tag{22}$$

In the following, let $\mathbf{b}_t = \|\mathbf{Y}_t u\|_2^2$, and define $\mathbf{b}_t' = \mathbf{b}_t - \mathrm{E}[\mathbf{b}_t]$, and $\mu = \frac{2\alpha n\ln(2n/k)}{dk}$. First, note that

$$\mathrm{E}[\mathbf{b}_t] = \int_0^\infty \Pr[\mathbf{b}_t > \xi]d\xi = \mu + \int_\mu^\infty \Pr[\mathbf{b}_t > \xi]d\xi$$

$$\le \mu + \int_\mu^\infty \frac{2n}{k}\cdot\exp\left(-\frac{dk}{2\alpha n}\cdot\xi\right)d\xi$$

$$= \mu + \underbrace{\left[-\frac{2n}{k}\cdot\frac{2\alpha n}{dk}\cdot\exp\left(-\frac{dk}{2\alpha n}\cdot\xi\right)\right]_\mu^\infty}_{\le \frac{2\alpha n}{dk}}$$

$$\le 2\mu.$$

In addition, it holds that

$$\mathrm{E}\left[\exp\left(\frac{|\mathbf{b}'_t|}{8\mu}\right)\right] = \int_0^\infty \Pr\left[\exp\left(\frac{|\mathbf{b}'_t|}{8\mu}\right) > \xi\right]d\xi$$

$$\leq 3/2 + \int_{3/2}^\infty \Pr[|\mathbf{b}'_t| > 8\mu\ln(\xi)]d\xi$$

$$= 3/2 + \int_{3/2}^\infty (\Pr[\mathbf{b}_t > 8\mu\ln(\xi) + \mathrm{E}[\mathbf{b}_t]] + \Pr[\mathbf{b}_t < -8\mu\ln(\xi) + \mathrm{E}[\mathbf{b}_t]])d\xi$$

$$\leq 3/2 + \int_{3/2}^\infty \left(\Pr[\mathbf{b}'_t > 8\mu\ln(\xi)] + \underbrace{\Pr[\mathbf{b}_t < -8\mu\ln(3/2) + 2\mu]}_{0}\right)d\xi$$

$$\leq 3/2 + \int_{3/2}^\infty \frac{2n}{k} \cdot \exp\left(-\frac{dk}{2\alpha n} \cdot 8\mu\ln(\xi)\right)d\xi$$

$$= 3/2 + \int_{3/2}^\infty \xi^{-8}d\xi$$

$$\leq 2.$$

Namely, $\mathbf{b}'_t$ is a Sub-Exponential random variable (Definition B.7) with $\|\mathbf{b}'_t\|_{\psi_1} \leq 8\mu$. Since $\{\mathbf{b}'_t\}$ are independent, each has zero mean, we obtain by Fact B.8 that

$$\forall \xi \geq 0: \quad \Pr\left[\sum_{t=1}^k \mathbf{b}_t \geq 2k\mu + \xi\right] \leq \Pr\left[\sum_{t=1}^k \mathbf{b}'_t \geq \xi\right] \leq 2\exp\left(-\Omega\left(\min\left(\frac{\xi^2}{k64\mu^2}, \frac{\xi}{8\mu}\right)\right)\right).$$

We now take $\xi = c \cdot \mu d\ln(k/\alpha) \geq c\alpha \cdot \log(k/\alpha)\log(2n/k) \cdot \frac{n}{k}$ for large enough constant $c$. Recall that by our assumption on $d$ it holds that $\xi \geq 4k\mu$. Hence

$$\Pr\left[\sum_{t=1}^k \|\mathbf{Y}_t u_t\|_2^2 \geq \xi\right] \leq \Pr\left[\sum_{t=1}^k \mathbf{b}_t \geq 2k\mu + \xi/2\right]$$

$$\leq \exp(-15d \cdot \ln(k/\alpha)),$$

$$\leq \exp(-d \cdot \ln(3k/\alpha) - 10),$$

where the second inequality holds assuming that $c$ is large enough. This concludes the proof of Equation (21) and therefore the proof of the lemma. $\qquad\square$

## E  Broader Impact

In this work we develop algorithms that maintain the differential privacy of the input points. When the points represent individuals, our work helps to maintain the privacy of those individuals.

