# OpenReview forum: "On Differentially Private Subspace Estimation in a Distribution-Free Setting"
_NeurIPS.cc/2024/Conference — NeurIPS 2024 poster_

### Official Review · Reviewer_RoTv · 2024-07-07

**Soundness:** 3
**Presentation:** 3
**Contribution:** 3
**Rating:** 6
**Confidence:** 3

**Summary:**

The paper tackles the challenge of high costs in private data analysis due to the curse of dimensionality, despite many datasets having an underlying low-dimensional structure. It builds on prior work by introducing measures based on multiplicative singular-value gaps to quantify how "easy" a dataset is for private subspace estimation. The authors provide new bounds and a practical algorithm that can estimate subspaces with a number of points independent of the dimension, showing improved performance in high-dimensional settings compared to previous approaches.

**Strengths:**

1.The problem of the paper is well-motivated and the authors give both upper bounds and lower bounds.
2.They also provide experimental results.
3. The writing is clear.
4. The literature review is relatively comprehensive.

**Weaknesses:**

There are some gaps between their upper bounds and lower bounds.

**Questions:**

1. Can your approaches work on real-world datasets?
2. Can you further improve the gap between upper bounds and lower bounds?

**Limitations:**

There are some gaps between their upper bounds and lower bounds.

---

> ### Author Rebuttal · Authors · 2024-08-05
>
> We thank you for the positive review.
>
> Regarding your questions:
>
> 1. We made an effort to reduce the constants so that our algorithm can achieve high accuracy using a reasonable amount of points. But as we mentioned in Section 5, our method is only effective for instances that are very close to a low-dimensional subspace, which seems unavoidable in the high dimensional regimes due to our lower bounds. We currently don’t know if interesting real-world datasets, like the gradients during training of large neural networks, have such a strong property that can be exploited for DP. We believe that it is an interesting research direction to understand the connection between the network and input structures to the phenomena of gradients near a low-dimensional subspace (Section 1.1). The work of Gur-Ari et al. [2018] (CoRR abs/1812.04754) makes us optimistic that in some cases it is possible to see this strong property, but we leave this direction for future work.
>
> 2. We currently don’t know how to improve the gap between the bounds. We conjecture that our lower bound for strong estimators is tight because it generalizes the tight lower bound of Dwork et al. 2014 which only holds for instances with a multiplicative singular-value gap above some specific constant. Yet, it is important to note that prior to our work, the gap between the lower bound of Dwork et al. 2014 and the upper bound of Singhal and Steinke 2021 was far beyond quantitative because the settings were very different. So we believe that our work, although it left quantitative gaps, makes a significant qualitative step towards understanding this problem better.

---

### Official Review · Reviewer_ivMD · 2024-07-12

**Soundness:** 3
**Presentation:** 2
**Contribution:** 3
**Rating:** 7
**Confidence:** 3

**Summary:**

This paper addresses the challenge of private subspace learning without assuming that the dataset follows a Gaussian distribution. They propose new measure on the hardness of subspace learning problem based on the ratio between the $k^{th}$ and the remaining eigenvalues, or between the $k^{th}$ and the $(k+1)^{th}$ eigenvalues. They derive upper and lower bounds on the utility guarantees for private subspace estimation for datasets that meet these criteria, showing that the bounds can be dimension-independent under suitable conditions. However, these conditions depend on some additional parameters, e.g. $\lambda$, that might need to be carefully selected. Also, the paper presents an implementable algorithm that achieves the derived upper bounds.

**Strengths:**

- The paper contributed to remove a restrictive assumption in previous works. Previous research on private subspace estimation has derived dimension-independent sample complexity for Gaussian-distributed data with large multiplicative eigen-gaps. This paper removes the Gaussianity assumption, and proves that dimension-independent sample complexity can still be achieved for easier datasets.
- The proposed measure of "easiness" of the datasets for subspace estimation, based on multiplicative eigen-gap, is simple and intuitive. Their derived bounds on sample complexity require prior knowledge of this measure, however, they show that the bounds are hardly affected when this measure is agnostic.
- The paper has practical relevance due to its realistic assumptions and the inclusion of an implementable algorithm for private subspace estimation.

**Weaknesses:**

The description of some parameters is vague, which hinders the understanding of the significance of the sample complexity bounds. For example, while $\lambda$ is an important parameter that appears in the sample complexity bounds, it is not defined in Definition 1.3 or in the text above it.

**Questions:**

Can you explain the parameter $\lambda$? Is it a parameter that can be freely selected, or is it intrinsically determined by the property of the dataset or the algorithm?

**Limitations:**

I don't see any significant limitations.

---

> ### Author Rebuttal · Authors · 2024-08-05
>
> We thank you for the positive review.
>
> Regarding your question:
> We are interested in DP algorithms that have the property that if they get "easier" inputs, they achieve "better" accuracy. The parameter $\lambda$ captures the connection between "easiness" and "accuracy" (i.e., it is a property of the algorithm).  In the setting of subspace estimation, if we aim for an $\alpha$-useful projection and we have an $\gamma$-easy dataset, then we would need to use an $\lambda$-estimator for $\lambda \leq \alpha/\gamma$.
>
> We decided to use this parameter since it captures (out of these three parameters) what mostly affects the sample complexities and our upper bounds' design (the number of subsets in the sample and aggregate), and is similar to the recent formulations of Peter et al. 2023. But this is just a matter of taste, and alternatively, we could avoid this parameter and formulate our results with $\alpha$ and $\gamma$.

---

> > ### Comment · Reviewer_ivMD · 2024-08-08
> >
> > Thanks for your clarification. It's very helpful for my understanding, and I increase my score accordingly.

---

### Official Review · Reviewer_UdSk · 2024-07-14

**Soundness:** 4
**Presentation:** 2
**Contribution:** 3
**Rating:** 7
**Confidence:** 3

**Summary:**

This work studies the problem of differentially private dimension reduction without dependence on the ambient dimension d. While this is known to be impossible generally, this work gives (distinct) necessary and sufficient conditions on gaps between the kth singular value of the data matrix and its subsequent singular values for privately identifying a k-dimensional subspace that captures the data with minimal distortion. The theoretical results are empirically verified, demonstrating that for datasets that are very close to a k-dimensional subspace, for small k, that the algorithms demonstrating sample upper-bounds do outperform existing, general DP subspace estimation techniques.

**Strengths:**

This work makes significant progress on an interesting problem, and the contributions could potentially have applications to notably improving efficiency of the broadly useful DP-SGD algorithm (in cases where gradients approximately lie in low-dimensional subspaces).

**Weaknesses:**

The presentation of this work could be improved. The reader's first introduction to what gamma-easiness means and how it eliminates dimension dependence comes in a footnote in page 2. I think that given the centrality of easy instances to this work, it would be good to dedicate a few sentences early on to give the reader a more precise understanding of what gamma-easiness means generally and how we should think of how gamma will relate to other parameters of a problem.

I also found the lower-bounds overview exceptionally hard to follow. These results are very technical and make use of prior work on fingerprinting codes and DP, so it does seem challenging to give an overview that is both sufficiently high level for the page limits and also illuminating, but I think it should be possible to make the overview more modular, and in that way easier to follow.

Typos/Suggested edits:

Section 1.5.2

“combination between generating” -> “combination of generating”

“However, their result strongly rely” -> “However, their results strongly rely”

“top-singular vector” -> “top-singular vectors”

1.5.3

“which have the property that we seek for” -> “which have the property we seek”

1.7

“Additional related work appear at Appendix A. Notations, …” -> “Additional related work appears in Appendix A. Notation, …”

2

“that arbitrary partition” -> “that arbitrary partitions”

“expected Forbenius”

“The first one simply treat each matrix” -> “treats”

“who privately estimate” -> “which privately estimates”

“asymptotical” -> “asymptotic”

Footnote 4 on page 6 defines the Frobenius norm, but it’s already been referred to in Definition 1.2, so could be good to move it earlier, also Frobenius is mispelled in the footnote.

3

Paragraph 4 of Section 3, second line, I think sigma_i should be sigma_i^2

4

“that are approximately lie” -> “that approximately lie”

5

“Yet, it still left open to close the gap” -> “Yet, closing the gap is still left open”

“Forbenius”

“via a private subspace estimation” -> “via private subspace estimation”

Appendix B

Proof of proposition B.12 “$Pi_1, Pi_2$ is holds” -> “it holds”

Appendix C.1

“that uses an oracle access” -> “that uses oracle access”

“we obtain that the projection matrices” -> “the projection matrices”

“simply treat the matrices … and compute” -> “simply treats … and computes”

Algorithm C.1

“Randomly split X intro t” -> “Randomly split X into t”

**Questions:**

The experiments all involve very small choices of k. Does this reflect an appropriate choice of constant dimension for the intended applications, computational limits, or performance limitations?

**Limitations:**

Yes, the authors have addressed the limitations of their work.

---

> ### Author Rebuttal · Authors · 2024-08-05
>
> We thank you for the positive and thorough review. Your editorial comments are very helpful and we will address them in the next version. In the following, we would like to respond to specific points you make.
>
> Regarding the weaknesses:
> In the next version, we will dedicate a few more sentences to clarify the connection between gamma and the other parameters (usefulness, and lambda), and we will try to improve the overview of our lower bounds.
>
> Regarding the question:
> Our implementation has time and space complexities of $\tilde{O}(dn+dk)$.
> In our experiments, we use $n = 250\cdot k$ points, so up to the hidden constant factors, the complexities are $\tilde{O}(dk)$. Since we focused on large values of $d$, we decided to use small values of $k$ to reduce the total running time (which involved 30 repetitions for each graph point), but there is no problem with using larger values of $k$.
> We remark that we did not try to optimize the running time. In particular, we did not use the fact that the heavy parts of our algorithm can be parallelized.

---

> > ### Comment · Reviewer_UdSk · 2024-08-12
> >
> > Thank you for answering my questions regarding parameter choices and for committing to improve some presentation issues. I will keep my score.

---

### Official Review · Reviewer_T3Ub · 2024-07-17

**Soundness:** 3
**Presentation:** 3
**Contribution:** 3
**Rating:** 7
**Confidence:** 4

**Summary:**

The paper studies private PCA under the assumption that the singular values of the data matrix shows multiplicative decay. It is a problem that was initiated by Steinke-Singhal and has not seen much improvement since that work. This works provides a more thorough study of this problem.

**Strengths:**

PCA is one of the most important problem in modern data analysis and ML. Unfortunately, with privacy constraints, it requires a lot of data samples to perform PCA with any reasonable accuracy. This has led researchers to look at instances where PCA can be easy. Most of these works rely on the assumption that the singular values of the data matrix has a particular behavior. This work extends on those line of work.

It provides nice upper bound on various assumptions regarding the multiplicative decay of the singular values. The upper bound result follows the framework presented in Steinke-Singhal. The lower bound to me are the more interesting one, which is based on recent ideas of Peter et al.

**Weaknesses:**

The lower and upper bound do not match. There is a large gap between the two. One major focus in the area of PCA is to get a bound under these assumptions such that when $k \to d$, we recover the optimal rate. For example, this was the motivation behind Mangroubi-Vishnoi. Unfortunately, under that assumption, the upper bound is way worse due to its dependence on $k^2 \sqrt{d}$ and $k^3d$, respectively. Due to that reason, I believe that the lower bound is tight.

**Questions:**

Do the authors perceive a way to improve their upper bound to get better dependence on $k$?

**Limitations:**

The quadratic and cubic dependence on $k$.

---

> ### Author Rebuttal · Authors · 2024-08-05
>
> We thank you for the positive review.
>
> Regarding your question. We currently don’t know how to improve the upper bounds, and we agree with your intuition that the lower bounds seem closer to the truth. One supporting evidence for this intuition is that our lower bound for strong estimators generalizes the tight lower bound of Dwork et al. 2014 that only holds for instances that have a multiplicative singular-value gap above some specific constant. Yet, it is important to note that prior to our work, the gap between the lower bound of Dwork et al. 2014 and the upper bound of Singhal and Steinke 2021 was far beyond quantitative because the settings were very different. So we believe that our work, although it left quantitative gaps, makes a significant qualitative step towards understanding this problem better.

---

### Decision · Program_Chairs · 2024-09-25

**Decision:**

Accept (poster)

**Comment:**

This paper studies the problem of differentially private subspace estimation. By assuming the multiplicative decay singular values, the author provide through studies of this problem.

All the reviewers agree that the paper provides interesting results and makes significant contributions to the field. In light of this, I recommend the paper for acceptance.